# An extended reconstruction of human gut microbiota metabolism of dietary compounds

Telmo Blasco [1,2], Sergio Pérez-Burillo [3,7], Francesco Balzerani[1,2,7], Daniel Hinojosa-Nogueira [3,7], Alberto Lerma-Aguilera [4,5,7], Silvia Pastoriza [3], Xabier Cendoya [1,2], Ángel Rubio[1,2], María José Gosalbes[4,5], Nuria Jiménez-Hernández[4,5], M. Pilar Francino [4,5✉], Iñigo Apaolaza [1,2✉], José Ángel Rufián-Henares [3,6✉] & Francisco J. Planes [1,2✉]

Understanding how diet and gut microbiota interact in the context of human health is a key question in personalized nutrition. Genome-scale metabolic networks and constraint-based modeling approaches are promising to systematically address this complex problem. However, when applied to nutritional questions, a major issue in existing reconstructions is the limited information about compounds in the diet that are metabolized by the gut microbiota. Here, we present AGREDA, an extended reconstruction of diet metabolism in the human gut microbiota. AGREDA adds the degradation pathways of 209 compounds present in the human diet, mainly phenolic compounds, a family of metabolites highly relevant for human health and nutrition. We show that AGREDA outperforms existing reconstructions in predicting diet-specific output metabolites from the gut microbiota. Using 16S rRNA gene sequencing data of faecal samples from Spanish children representing different clinical conditions, we illustrate the potential of AGREDA to establish relevant metabolic interactions between diet and gut microbiota.

[1] Tecnun, University of Navarra, San Sebastián, Spain. [2] Biomedical Engineering Center, University of Navarra, Campus Universitario, Pamplona, Navarra, Spain. [3] Departamento de Nutrición y Bromatología, Instituto de Nutrición y Tecnología de los Alimentos, Centro de Investigación Biomédica, Universidad de Granada, Granada, Spain. [4] Área de Genòmica i Salut, Fundación para el Fomento de la Investigación Sanitaria y Biomédica de la Comunitat Valenciana-Salud Pública, Valencia, Spain. [5] CIBER en Epidemiología y Salud Pública, Madrid, Spain. [6] Instituto de Investigación Biosanitaria ibs.GRANADA, Universidad de Granada, Granada, Spain. [7] These authors contributed equally: Sergio Pérez-Burillo, Francesco Balzerani, Daniel Hinojosa-Nogueira, Alberto Lerma-Aguilera. ✉email: francino_pil@gva.es; iaemparanza@tecnun.es; jarufian@ugr.es; fplanes@tecnun.es

Understanding how diet and gut microbiota interact in the context of human health is a key question in personalized nutrition[1]. Compounds derived from the diet affect the abundance of different species present in the gut microbiome, which, on the other hand, release key metabolites and signals that regulate host health. The relevance of this interaction is supported by an increasing body of literature showing that the beneficial effect of dietary interventions in different clinical conditions is associated with specific signatures of the gut microbiota[1–3].

Given the complex molecular events implied in this question, the development of computational models, driven by meta-omics data, constitutes a major task in systems biology[4,5]. In particular, the integration and analysis of genome-scale metabolic models of different bacterial species that are present in the human gut microbiota have received much attention[6]. Thanks to the tremendous effort in the last years to generate high-quality computational platforms for metabolic reconstruction[7–10], extensive microbial community models of the human gut microbiome are now available. In particular, AGORA constituted the first large effort in the literature, involving 818 species present in the human gut microbiota[11].

These network-based community models, which integrate the metabolic capabilities of different bacterial species in the gut microbiome, can be analyzed via constraint-based modeling (CBM)[12–14]. This approach is promising in personalized nutrition and could help in elucidating how different microbial species in the human gut exploit and transform nutrients derived from the diet and in systematically designing effective dietary strategies when the gut microbiome is dysregulated. For example, AGORA has been already applied to predict dietary supplements for Crohn's disease[15]. Using a similar approach, we predicted the effect of solid diet on the gut microbiota metabolism of infants[16]. Despite these early attempts, genome-scale metabolic models of the gut microbiota are still in their infancy and further developments are required to make them into a practical tool in personalized nutrition.

A major issue of current metabolic reconstruction platforms is the limited information about metabolic pathways of dietary compounds that are transformed by the gut microbiota. AGORA only includes 99 out of 650 dietary compounds included in i-Diet, a commercial software for personalized nutrition (http://www.i-diet.es). A similar result was found for CarveMe[8], a more recent metabolic reconstruction platform, which involves 92 dietary compounds from i-Diet in the 5587 bacterial metabolic models reported. In addition, universal metabolic databases, such as the Model SEED[7], on which reconstruction platforms rely for gap filling, are incomplete and include metabolic capabilities of species that are not present in the human gut. Overall, these limitations restrict the scope of CBM approaches to predict the interaction between diet and gut microbiota.

In this article, using a combination of bioinformatic tools, metabolic databases, and literature, we extend AGORA and substantially improve the coverage of gut microbial metabolism of dietary compounds. Particularly, we include degradation pathways of 209 dietary compounds (not present in AGORA), from which 179 are phenolic compounds, a family of metabolites highly relevant for human health and nutrition that are mainly transformed by the gut microbiota[17,18]. Our reconstruction, called AGREDA (AGORA-based REconstruction for Diet Analysis), is thus more amenable to analyze the role of the human gut microbiota in diet metabolism.

To illustrate our contribution, we first show that AGREDA provides a more complete connection than AGORA to the nutritional composition of 20 typical recipes of the Mediterranean diet. In addition, using 16S rRNA gene sequencing data, we apply AGREDA to predict output microbial metabolites from in vitro fermentation of lentils with feces of Spanish children representing different conditions: normal weight, obesity, allergy to cow's milk, and celiac disease. We provide experimental validation and compare AGREDA with AGORA using both targeted and untargeted metabolomic strategies. Finally, for the same children, we assess the metabolic interaction between the 20 recipes mentioned above and the gut microbiota, finding a substantially higher number of significant associations in AGREDA than in AGORA. In conclusion, AGREDA addresses the necessary intersection between human nutrition, metagenomics, and computational modeling to advance towards personalized nutrition.

## Results

We present a new metabolic reconstruction of the human gut microbiota that is focused on covering significant gaps in the degradation pathways of dietary compounds into terminal downstream metabolites. Our reconstruction follows a mixed-bag community model[19] (see "Methods" section), where reactions from different organisms are merged into a single compartment. This strategy reduces the size of reconstruction and computation time and has been proved accurate and effective for analyzing the metabolic capabilities of the gut microbiota as a whole[20,21], which is the aim of our study, focused on predicting output metabolites from the gut microbiota in different conditions. However, we store for each reaction its taxonomic assignment (Fig. 1), which allows us to model the interaction between different species and integrate meta-omics data (see "Methods" section).

We started from AGORA[11], which includes 818 reconstructions of bacterial species present in the human gut microbiota. Following the mixed-bag network strategy (see "Methods" section), we removed the boundaries between the different species and deleted duplicated reactions, obtaining 2473 metabolites and 5312 different reactions, together with their taxonomic assignments. Henceforth, this summarized network is referred to as AGORA.

We then built a universal metabolic network based on the Model SEED[7] database and manually curated literature knowledge. Through their Enzyme Commission (EC) numbers (if available), reactions were annotated to species present in AGORA using different bioinformatics tools and metabolic databases (Fig. 1 and "Methods" section). This universal network was consistently integrated with the reactions and metabolites from AGORA.

Next, we applied a gap-filling algorithm to include in our reconstruction the maximum number of dietary compounds and their degradation pathways. This step was based on FastCoreWeighted, included in the COBRA Toolbox[22,23] (see "Methods" section). All the reactions extracted from the universal database to fill existing gaps included taxonomic assignment to species in AGORA, which provides support for the predicted pathways.

Finally, a single-species analysis was applied to the resulting metabolic model in order to identify blocked reactions and discard pathways requiring transport reactions with limited evidence in public databases (see "Methods" section). We also conducted similar quality checks to those originally performed with AGORA: aerobic and anaerobic growth in different growth media, as well as carbon source uptakes and fermentation product secretions for different species (Supplementary Fig. 1). Our final reconstruction is called AGREDA (Supplementary Data 1).

AGREDA adds to AGORA 809 reactions and 320 metabolites, from which 209 are input dietary compounds from i-Diet not included in AGORA. We improve the coverage of a wide number

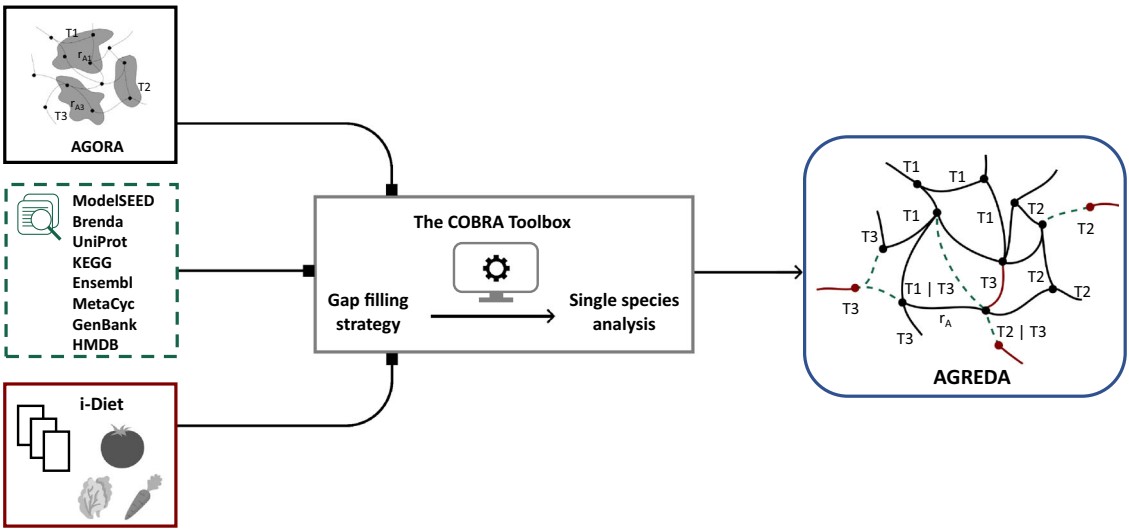

**Fig. 1 Summary of the reconstruction pipeline.** First, AGORA reconstructions[11] (black) are merged into a single compartment. Duplicated reactions were deleted but the taxonomic assignment was kept. For example, the same reaction $r_A$ in taxon 1 (T1) and taxon 3 (T3) in AGORA, $r_{A1}$ and $r_{A3}$, were converted to only one reaction in AGREDA and its associated Boolean rule, $T_1|T_3$, which we term Taxonomy-Reaction (TR) rule. Next, the Model SEED[7] reactions (green) are annotated to AGORA species through EC number information (see "Methods" section). Then, metabolites provided by i-Diet and manually curated literature knowledge are integrated with AGORA and the Model SEED (maroon). Finally, gap-filling techniques and single-species analysis, based on the Cobra Toolbox[22,23], are applied to derive AGREDA.

of metabolic subsystems (Supplementary Table 1), particularly those involved in different families of phenolic compounds (see below). Other relevant metabolic tasks of the gut microbiota were also extended, including the biosynthesis of carotenoids, such as beta-carotene, the precursor of vitamin A[24], amino acid metabolism, particularly the secretion of citrulline[25], alternative pathways for the biosynthesis of GABA[26], and caffeine metabolism[27], among others.

All the reactions in AGREDA have taxonomic annotation to species present in AGORA. Full details can be found in Supplementary Data 2. Figure 2a shows the number of reactions and metabolites related to each species grouped by the respective phyla. It can be observed that all phyla contain a higher number of metabolites in AGREDA than in AGORA. Specifically, each phylum in AGREDA contains on average 210 reactions and 120 metabolites more than in AGORA. As a result, the metabolic differences among species are captured substantially better in AGREDA than in AGORA, according to the average metabolic distance across species calculated with the Jaccard's distance (0.58 vs 0.48, respectively, Supplementary Fig. 2).

An important set of metabolites included in AGREDA is that of phenolic compounds. These compounds are widespread in the vegetal kingdom, where they act as a defensive system against external aggressions and have been pointed out to be responsible for many of the health benefits of vegetable consumption. AGREDA covers a very wide range of phenolic compounds, from the simpler ones (benzoic and hydroxycinnamic acids) to the more complex (proanthocyanidins), with all families represented (Fig. 2b). Overall, AGREDA added degradation pathways for 179 phenolic compounds present in the diet, significantly improving the coverage of AGORA, which only contained 15 phenolic compounds.

The daily intake of phenolic compounds is high in comparison to that of most micronutrients, since they are especially abundant in highly consumed food items such as tea or coffee (specially rich in cinnamic acids and flavan-3-ols) and fruits, vegetables, and legumes (wide range of different flavonoids)[28]. In the case of Spanish children, our group has estimated an average intake of phenolic compounds of 2079 mg/day[29]. However, the phenolic compounds are barely absorbed in the small intestine and reach the gut microbiota where they are metabolized by organisms belonging to different phyla, usually into smaller molecules that are more easily absorbed in the large intestine[30]. Therefore, the benefits of most phenolic compounds are actually exerted by their output metabolites, hence the importance of being able to define their microbial metabolization[31]. Figure 2c shows the degradation capabilities of different phyla for three families of phenolic compounds: flavanones, benzoic acids, and hydroxycinnamic acids.

In addition to phenolic compounds, we improved the coverage of other relevant families of metabolites present in the diet, ranging from carbohydrates, carotenoids, fats, minerals, phytosterols, vitamins, and xanthines (Fig. 2d). Overall, AGREDA included 30 dietary compounds from these families that are not present in AGORA.

In order to assess the improvement that AGREDA represents over AGORA for the purpose of assessing the effects of the different diets on the gut microbiota metabolism, we selected 20 representative recipes and employed i-Diet to calculate the nutrients present in each of them (Supplementary Data 3). As shown in Fig. 3a, only approximately half of the nutrients of each recipe that are captured by AGREDA are also present in AGORA. In addition, the heatmaps in Fig. 3b represent the dissimilarity (Jaccard's distance) among the sets of nutrients present in each recipe captured by AGORA and AGREDA, respectively. We observe that AGREDA performs better at capturing the differences between recipes, according to their input nutritional composition. A similar result was obtained when comparing AGREDA with CarveMe (Supplementary Fig. 3). We can, therefore, conclude that AGREDA provides us with a more comprehensive tool to assess the effect of the gut microbiota on diet metabolism.

**Prediction of output metabolites from in vitro fermentation of lentils with children's feces.** A commercial Spanish recipe of boiled lentils was fermented in vitro with fecal inocula from

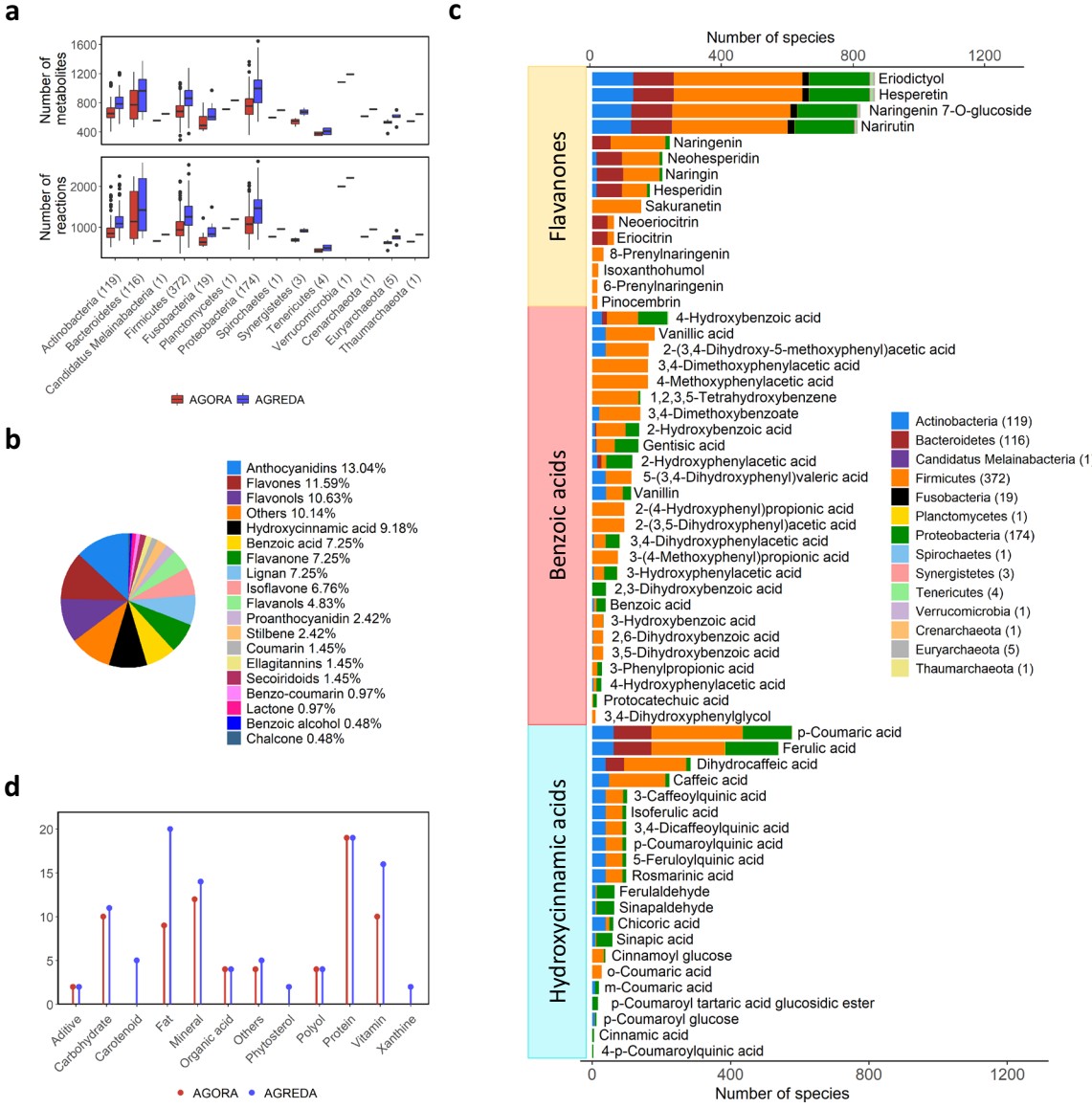

**Fig. 2 Main features of AGREDA. a** Boxplots depict for each phylum the number of metabolites and reactions of its associated strains in AGORA (red) and AGREDA (blue). Bottom and top of the boxes denote the first and third quartiles, respectively, and whiskers represent the values within 1.5 interquartile range above and below the box. Center line represents the median value. The number of strains per phylum is shown in brackets. Blocked reactions and metabolites were excluded from metabolic models derived from AGORA and AGREDA; **b** Distribution of the 179 phenolic compounds added by AGREDA separated in 19 families. **c** Degradation capabilities for three families of phenolic compounds present in AGREDA. The total number of strains in each phylum is reported in brackets; **d** Other families of compounds in the diet included in AGREDA and AGORA.

children belonging to four different clinical conditions, i.e., normal weight, obesity, allergy to cow's milk, and celiac disease. Seven inocula were prepared with the fecal samples from lean, obese, and celiac children, while six were prepared with those from children allergic to cow's milk, for a total of 27 fermentations. The taxonomic composition of the microbiota present in the different fermentations was measured via 16S rRNA gene sequencing (see "Methods" section).

In order to assess the role of the gut microbiota as a whole during the fermentation of lentils for each of the different conditions detailed above, we contextualized the reference AGREDA and AGORA models with the nutritional information of the lentils recipe from i-Diet and the taxonomic composition of the fecal inocula (see "Methods" section, Supplementary Data 3), obtaining 27 context-specific AGORA and 27 context-specific

AGREDA models. For each context-specific metabolic model, using flux variability analysis (FVA)[32], we determined the potential list of output microbial metabolites (by-products) that can be derived from the fermentation of lentils. On average, for each sample, AGREDA predicted 98 output metabolites that were not captured by AGORA.

Then we compared the predictive potential of the respective context-specific AGORA and AGREDA models in identifying the presence or absence of ten output microbial phenolic compounds derived from the fermentation of lentils (Fig. 4). For validation purposes, we used targeted metabolomics analysis in three inocula per clinical condition, for a total of 12 samples (see "Methods" section for details, Supplementary Data 3). Thus, we compared AGORA and AGREDA predictions in 120 cases (12 samples × 10 compounds; Fig. 4a).

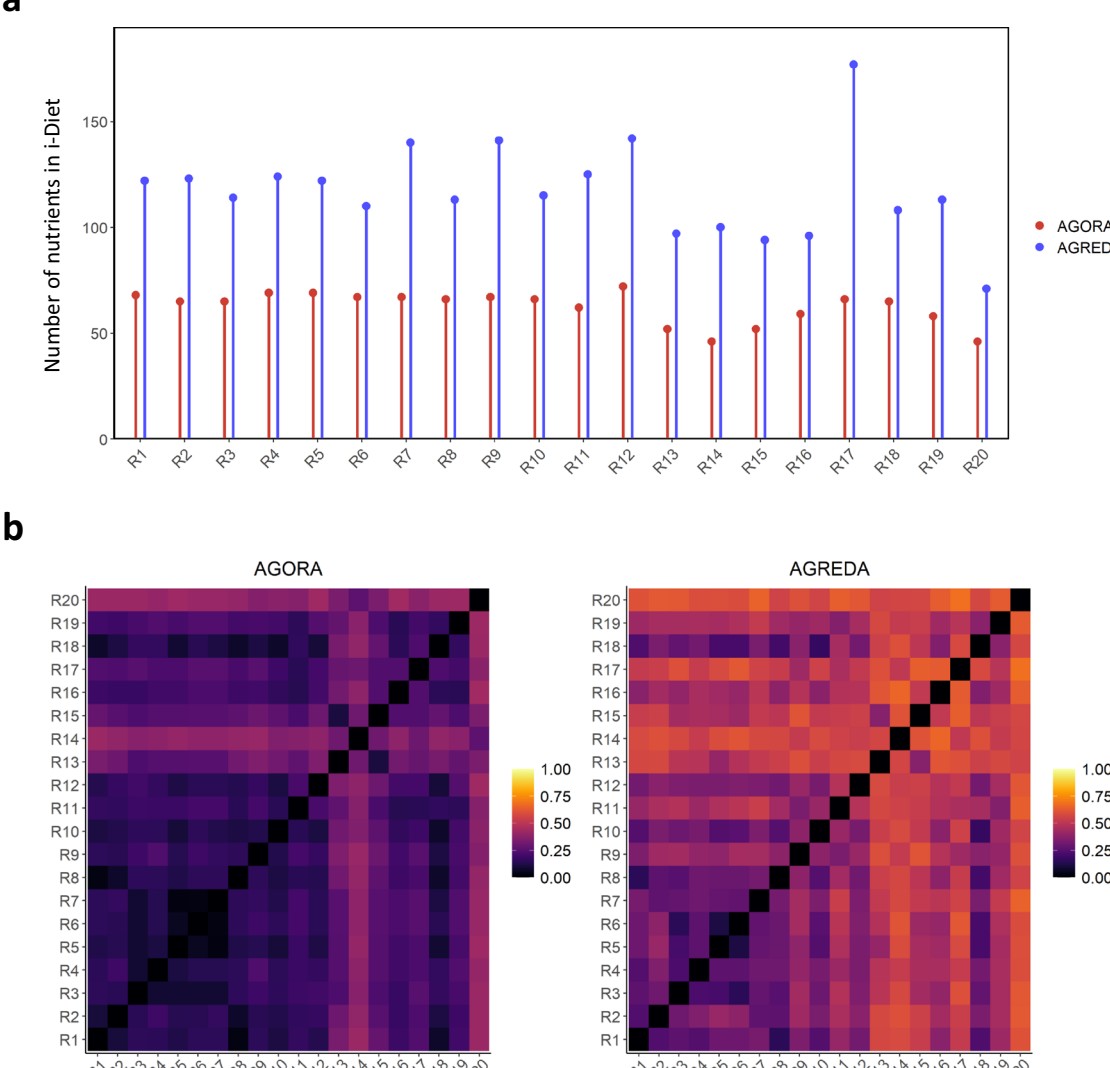

**Fig. 3 Nutritional composition of 20 representative recipes of the Mediterranean diet in AGREDA and AGORA. a** The number of input dietary nutrients that AGORA (red) and AGREDA (blue) capture for different recipes (R1, R2, ..., R20). Note that all the metabolites present in AGORA are also included in AGREDA. **b** Differences between the nutritional content of the recipes captured by AGORA and AGREDA, respectively. The Jaccard's distance between the compositions of the recipes is represented.

As summarized in Fig. 4b, we found that AGREDA correctly identifies a large fraction of positive cases, and thus the sensitivity of the AGREDA context-specific models is remarkably higher than that of the AGORA context-specific models: 0.728 vs 0.223, respectively. Despite the fact that the specificity of AGREDA is lower than that of AGORA (0.765 vs 0.941, respectively), AGREDA globally outperforms AGORA (accuracy: 0.733 vs 0.325, respectively). This difference between AGREDA and AGORA is highly significant (Fisher test $p$ value: $1.55 \times 10^{-4}$ vs 0.1892, respectively). We repeated the same analysis for CarveMe; however, it was not able to predict most of the output metabolites and the results are poorer than those of AGORA (Supplementary Fig. 4).

We provided a more general experimental validation to the results derived from AGREDA by means of an untargeted metabolomics analysis in 24 out of the 27 fermentations considered above (see "Methods" section). In particular, we focused on output microbial metabolites with a different predicted outcome (presence/absence) between AGREDA and AGORA in at least one of the samples analyzed. This subset comprised 135 output metabolites; however, we discarded those not identified in any of the 24 samples analyzed, possibly due to the lack of sensitivity of the metabolomic approach adopted. Thus, we reduced the analysis to 105 output metabolites, among which 19 took part both in AGREDA and AGORA and 86 were only predicted by AGREDA.

The binary outcome obtained from metabolomic data for these 105 metabolites in the 24 samples analyzed can be found in Supplementary Data 3. In the case of 19 metabolites captured by both AGREDA and AGORA, we found a statistically significant association with metabolomic data in AGREDA but not in AGORA (Fisher test $p$ value: $1.876 \times 10^{-5}$ vs 0.1324, respectively; Supplementary Table 2). On the other hand, in the case of 86 metabolites only predicted by AGREDA, we obtained an even higher statistical association (Fisher test $p$ value: $7.205 \times 10^{-6}$; Supplementary Table 3). We, therefore, conclude that the new metabolites and pathways included in AGREDA significantly improve our predictive capacity of gut microbiota metabolism as a whole and enable the detection of output metabolites not considered in AGORA.

**a**

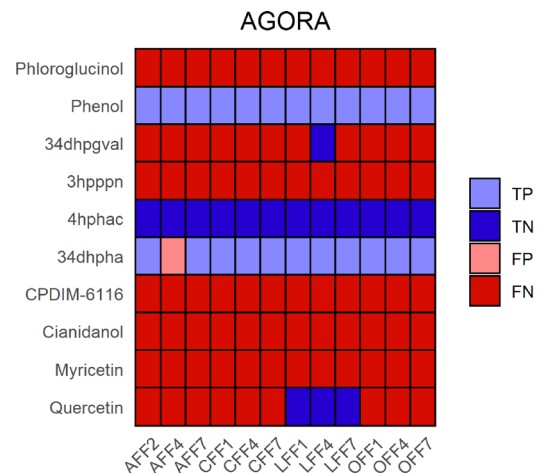
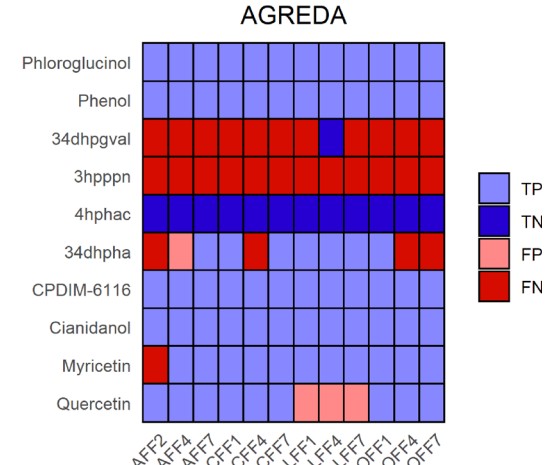

**b**

| Output metabolites | | AGORA | | AGREDA | |
|---|---|---|---|---|---|
| | | **+** | **−** | **+** | **−** |
| ***In vitro* experiment** | **+** | 23 | 80 | 75 | 28 |
| | **−** | 1 | 16 | 4 | 13 |

| | AGORA | AGREDA |
|---|---|---|
| **Sensitivity** | 0.223 | 0.728 |
| **Specificity** | 0.941 | 0.765 |
| **Accuracy** | 0.325 | 0.733 |
| **Fisher´s p-value** | 0.1892 | $1.55 \times 10^{-4}$ |

**Fig. 4 In vitro experimental comparison of the predictions by AGREDA and AGORA. a** Comparison of AGREDA and AGORA for predicting the presence (positive) or absence (negative) of ten output microbial compounds derived from the fermentation of lentils with children feces and measured with a targeted metabolomics approach. **b** Confusion matrix and statistical details of the comparison shown in **a**. Sensitivity was determined as TP/(TP+FN), specificity as TN/(FP+TN), and accuracy as (TP+TN)/(TP+TN+FP+FN). The reported Fisher's $p$ value was two-sided. 34dhpgval 5-(3′,4′-Dihydroxyphenyl)-gamma-valerolactone, 3hpppn 3-(3-hydroxy-phenyl)propionate, 4hphac 4-hydroxyphenylacetate, 34dhpha (3,4-dihydroxyphenyl) acetate, CPDIM-6116 dihydrocaffeic acid; "AFF2," "AFF4," and "AFF7" denote samples 2, 4, and 7 from children allergic to cow's milk, respectively; "CFF1," "CFF4," and "CFF7" denote samples 1, 4, and 7 from celiac children, respectively; "LFF1," "LFF4," and "LFF7" denote samples 1, 4, and 7 from lean children, respectively; "OFF1," "OFF4," and "OFF7" denote samples 1, 4, and 7 from obese children, respectively; TP true positives, TN true negatives, FP false positives, FN false negatives.

**In silico prediction of metabolic interaction between diet and children's microbiota**. We conducted a similar in silico analysis as the one presented above for the 20 recipes considered in Fig. 3. In this case, 16S rRNA gene sequencing data were obtained from fecal samples from the same children incubated with minimal growth medium (see "Methods" section). Input dietary compounds were fixed for each recipe and consisted of their associated nutrients in i-Diet, mucins, and compounds in the minimal medium (Supplementary Data 3).

Overall, we derived 27 context-specific AGREDA models for 21 different conditions, namely, 20 recipes and the minimal medium, which makes a total of 567 context-specific metabolic models. The same analysis was also accomplished with AGORA. As done in the lentils study discussed above, using FVA, we determined the potential list of output microbial metabolites that can be obtained from each of the different scenarios considered (Supplementary Data 3).

Based on the results of FVA, we analyzed the relevance of different recipes and clinical conditions in the production of output microbial metabolites. To that end, we built a logistic regression model for each output metabolite using each recipe and clinical condition as independent (explanatory) variables (see "Methods" section). We identified in AGREDA 151 output metabolites whose potential production is significantly affected by different recipes or clinical conditions (adj. $p$ value ≤ 0.05), among which 49 metabolites depend on the clinical conditions, 87 metabolites depend on diet (recipes), and 15 metabolites depend on both factors. The rest of the output microbial metabolites did not show any significant pattern for clinical conditions and diet.

Figure 5 shows heatmaps of predicted production of three different output metabolites across different recipes and clinical conditions. Figure 5a illustrates one of the cases where only diet affects the outcome, namely, only 2 recipes lead to the production of 4-methoxyphenylacetic acid. Similarly, Fig. 5b represents the situation where only the clinical conditions affect the outcome, particularly the production of sesamolin is only possible in the lean and celiac children considered in our analysis. Finally, both diet and clinical conditions are relevant in Fig. 5c, where pyrogallol can be produced with most recipes in lean and celiac children, while children who are obese or allergic to cow's milk require a more specific diet.

The same analysis was done in AGORA, finding that clinical conditions or diet are implied in the production of 43 output metabolites, a considerably lower number of significant associations than in AGREDA. In particular, we identified in AGORA 36 output metabolites affected by clinical conditions, 6 by diet, and 1 by both factors. The rest of the output metabolites did not show a significant pattern for clinical conditions and diet. In conclusion, AGORA presents a larger subset of output microbial metabolites whose production is dependent on diet and/or children's gut microbiota, which shows that AGREDA captures context-specific metabolic pathways not present in AGORA.

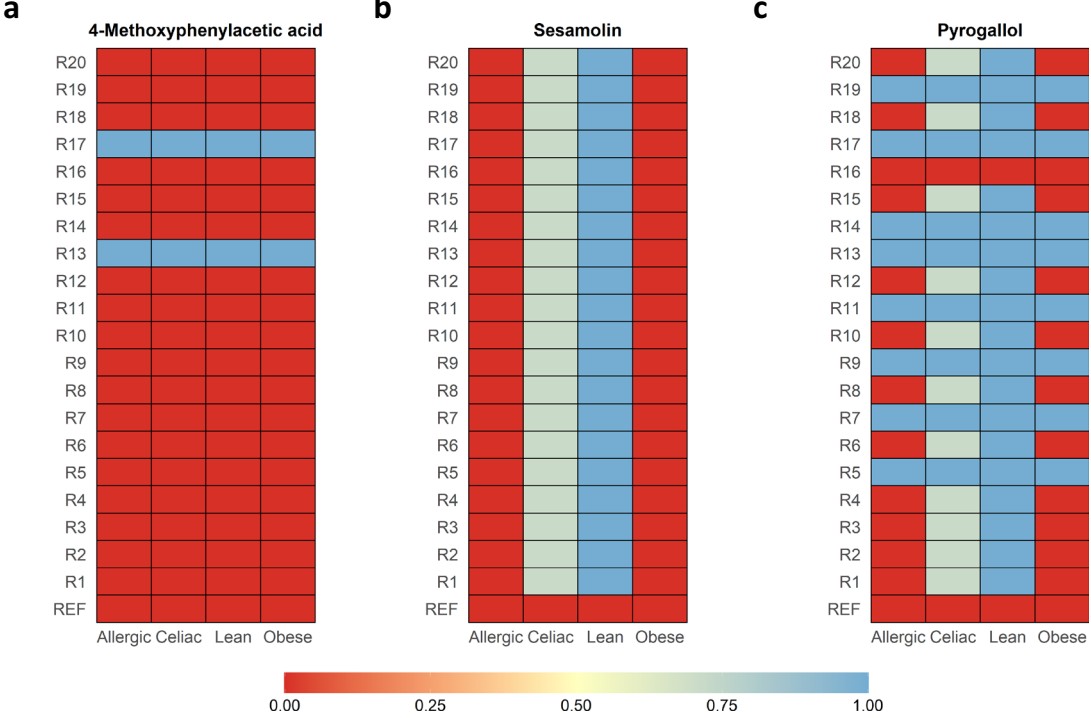

**Fig. 5 AGREDA-predicted production of three output microbial metabolites across different recipes and clinical conditions.** Each entry in the heatmap represents the proportion of samples of a particular clinical condition where **a** 4-methoxyphenylacetic acid, **b** sesamolin, and **c** pyrogallol are produced. Seven samples were used for celiac, lean, and obese children, while 6 samples were used for children allergic to cow's milk. The analysis was done for 20 different recipes (R1, R2, …, R20) and the minimal growth medium (denoted REF).

## Discussion

CBM constitutes a promising approach to investigate the interaction of diet and gut microbiota, as well as their impact on host health. In the past years, the number of high-quality genome-scale metabolic reconstructions of species present in the human gut has significantly increased, aiming to conduct a more comprehensive analysis of the gut microbiota metabolism. However, they need further developments to become a practical tool in the area of personalized nutrition, since a large variety of key nutrients present in the diet are not considered in these reconstructions. This limitation could substantially impair our study of the interplay between diet and gut microbiota metabolism.

In this article, we directly address this relevant issue and extend AGORA[11], one of the largest repositories of metabolic reconstructions of species present in the human gut microbiome. In particular, we integrate AGORA with 209 dietary compounds included in i-Diet, a commercial nutritional software designed to elaborate optimal diets, collectively involving 809 new reactions and 320 new metabolites. Our reconstruction, termed AGREDA, was built through an exhaustive literature analysis and gap-filling algorithms using Model SEED[7] as universal database. For this task, we used different bioinformatic tools to integrate Model SEED and AGORA and avoided the use of reactions with limited evidence in the human gut microbiota. As a result, our proposed reactions in AGREDA include taxonomic annotation to species present in AGORA, which facilitates the analysis of their activity with 16S rRNA gene sequencing data.

In addition, our reconstruction pipeline included a single-species analysis in order to identify blocked reactions and discard pathways requiring transport reactions with limited evidence in public databases. With this step, AGREDA could be used to construct compartmentalized network community models. However, for the predictive analysis presented in the "Results"

section, aimed at identifying output metabolites produced by the entire gut microbiota in different conditions, we decided to follow a mixed-bag strategy. This was done to reduce the size of the community model and therefore the computation time of our simulations. Given our positive results, in line with other works in the literature, this simplification does not seem to affect much our predictions. In fact, we reproduced the case study of lentil fermentation using a compartmentalized network community model and found very similar results (Supplementary Tables 4 and 5) but at a substantially higher computational expense than that of the mixed-bag network community model (120 vs 5 min, respectively). This shows that the AGREDA mixed-bag approach is effective and accurate in predicting the global metabolic capabilities of the human gut microbiota.

With the inclusion of the degradation pathways of 179 phenolic compounds, AGREDA constitutes the largest effort in the literature to compile the metabolism of these compounds by the human gut microbiome. Despite our advance, there is substantial room for improvement, since AGREDA currently only includes 97 out of 372 metabolites detailed in Phenol-Explorer[33], the first comprehensive database of polyphenol contents in foods. Many of them are not annotated in universal metabolic databases, such as KEGG[34,35] or Model SEED, requiring new strategies to address this issue. In this direction, enzyme promiscuity methods constitute a promising approach to further complete degradation pathways of phenolic compounds.

Importantly, AGREDA provides a more comprehensive integration between diet and gut microbiota metabolism than AGORA, as shown in Fig. 3 for 20 different representative recipes, where a significantly higher coverage of their nutrient composition was obtained. This advance logically allows us to carry out a more complete analysis of output metabolites from the gut microbiota. This was illustrated in the case study of lentils, where AGREDA

showed higher accuracy than AGORA in predicting output microbial metabolites in different samples analyzed, according to both targeted and untargeted metabolomics data.

In addition, we applied AGREDA to assess metabolic differences in the way the gut microbiome of Spanish children degrades input dietary compounds from 20 different recipes. Despite the limited number of samples and recipes considered in our analysis, AGREDA identified a remarkably higher number of significant associations between diet, gut microbiota, and output microbial metabolites than AGORA. This emphasizes that AGREDA provides a more complete definition of diet-specific metabolic pathways present in the gut microbiota.

For these reasons, AGREDA opens new avenues to investigate the interaction between diet and gut microbiota with views to developing personalized nutrition programs. Output microbial metabolites derived from diet may not only reach the blood stream and affect host homeostasis but could also have an impact on the structure of the gut microbial community. In combination with the nutritional software that can design and propose diets based on their nutrient content, AGREDA can contribute to personalized nutrition by defining the optimal nutrients for modulating the metabolic output of an individual-specific gut microbiota toward healthier states. The regulation through diet of key output microbial metabolites in different clinical conditions is an essential part of personalized medicine and AGREDA constitutes a relevant step forward in this direction.

## Methods

**Universal biochemical reaction database**. We start from AGORA 1.03[11], which comprises manually curated metabolic models of 818 species of the human gut microbiome. In order to reduce the computation cost, we modeled the overall microbial community as a mixed-bag network strategy, where reactions from different organisms are merged into a single compartment. Different works have shown this approach to be useful to get insights from microbial community models[36] and, particularly for the assessment of interactions between the bacteria and the environment[19]. A mixed-bag network strategy is suitable for the work presented here, which aims at identifying the output metabolites that will be potentially synthesized by a given set of bacterial species present in a specific nutritional environment. A detailed example illustrating the mixed-bag network community model can be found in Supplementary Note 1.

Briefly, the following steps were performed for the creation of the microbial community model from AGORA. First, we modeled the interactions between the bacterial species present in AGORA by forcing the different metabolic networks to share the extracellular metabolites. As a consequence, scenarios where two species compete for a given nutrient or one species consumes a metabolite secreted by another species can be considered. Second, we removed the compartments and boundaries between the different species and deleted duplicated reactions (two different species could have the same reaction). Nevertheless, all the reactions include their corresponding taxonomic assignation, namely, for each reaction, the bacterial species that can carry it out are annotated. Altogether, we obtained a metabolic model including 2473 metabolites and 5312 reactions that explains the overall metabolic capacity of the microbial community. Note here that, due to the inclusion of the species interactions in the first step and of the taxonomic assignations of the reactions in the second one, the information of the interactions at the species level is not lost.

In order to extend the metabolism of dietary compounds by the gut microbiota, we integrated the information provided by AGORA with the Model SEED database[7], as well as with other metabolic databases and literature knowledge of gut microbiota metabolism, as we detail below.

We first downloaded Model SEED from the online portal (https://modelseed.org/), which involves 20,133 metabolites and 34,655 reactions. To minimize the inclusion of reactions from species not active in the human gut microbiome, we decided to annotate the EC numbers present in the Model SEED with the species present in AGORA. Note here that Model SEED does not incorporate Gene-Protein-Reaction rules, as available in AGORA; instead, it presents a broad functional annotation of reactions through EC numbers. Therefore, the integration of Model SEED with AGORA can be done through the taxonomic annotation of its EC numbers. We describe below the different strategies followed to carry out this task with existing genomic annotation tools and relevant metabolic databases.

Genome FASTA files from different species in AGORA were downloaded from GenBank[37] and Ensembl[38] through the NCBI taxonomy identifier and species name, respectively. These genomes were annotated using the myRAST software from the RAST Server[39], which outputs their protein-encoding genes and (if available) associated EC numbers. This information was incorporated into the

reactions present in Model SEED. In addition, from the KEGG database[34,35], we downloaded the list of EC numbers for 500 species present in AGORA. With this information, we could further annotate reactions in Model SEED without taxonomic information.

We also performed a manual annotation of reactions and EC numbers present in Model SEED. We found that several reactions that did not contain any EC number information in Model SEED were annotated in public databases, such as KEGG or MetaCyc[40]. Based on them, we extracted more reactions with enzymatic information and used again the myRAST software for taxonomic annotation. For the remaining EC numbers without taxonomic information, we manually looked for additional information in the KEGG, BRENDA[41], and UniprotKB[42] databases. After this process, we obtained a list of 3577 different EC numbers and 14,021 reactions in Model SEED that are related to at least one of the species in AGORA.

We noticed that some metabolites in Model SEED were involved in reactions under different names. Using both manual curation and chemoinformatic tools, we identified and deleted metabolites that were duplicated. In particular, we first extracted the InChI identifier for the metabolites in Model SEED (13,028 out of 20,133 metabolites), based on PubChem[43], the Human Metabolome Database[44], KEGG, and the RetroRules database[45]. We then conducted a similarity analysis with the RDKit package[46] and the Morgan (circular) fingerprint with radius 2[47]. Fingerprints with similarity 1 were obtained and manually checked. We removed 703 repeated metabolites and 1054 reactions from Model SEED.

In order to integrate AGORA and Model SEED, we performed an automatic search of the compound names in both sources and identified duplicated metabolites and reactions. Model SEED added to AGORA 17,820 metabolites and 32,409 reactions, including 12,459 reactions with taxonomic assignment.

In addition, we manually identified the list of metabolites from i-Diet present in Model SEED and found 221 that were not present in AGORA. We created an exchange reaction for each of these compounds and included them in our metabolic database. We also added 221 reactions and 19 metabolites from the existing literature on metabolism of phenolic compounds in the gut microbiota, including their biochemical reaction annotation (Supplementary Data 2). After this final step, our universal biochemical reaction database reached 20,376 metabolites and 38,048 reactions. Note here that 20,023 and 13,478 of these reactions do not have taxonomic and functional assignment, respectively.

**Gap-filling strategy**. The universal biochemical reaction database described in the previous subsection was used to extend AGORA and connect input dietary compounds to output microbial metabolites or intermediate metabolites for biomass production. To that end, we used the implementation of FastCoreWeighted available in the COBRA Toolbox[22,23]. This reconstruction algorithm requires the definition of a subset of reactions that must take part in the resulting network, termed core, and efficiently identifies the necessary reactions from a universal database to make the core functional. In addition, it allows us to penalize differently the inclusion of reactions from our universal database. Here we set a weight equal to 0 for reactions in the core, 0.1 for reactions with taxonomic assignment to species in AGORA, 50 for reactions without taxonomic assignment but with functional annotation (at least one EC number available), 100 for reactions without taxonomic and functional annotation, and 1000 for reactions manually assigned to plant metabolism. The logic of these weights is to penalize reactions for which there is no taxonomic or functional evidence in the species reported in AGORA, i.e., gap filling is therefore preferably done through reactions annotated to species found in AGORA and therefore in the human gut.

As we found dependencies between different nutrients from i-Diet, namely, some of them are interconnected as inputs and outputs, we ran FastCoreWeighted sequentially, updating the core at each iteration. In the first iteration (Iteration 1), the core included the reactions from literature and AGORA. Note here that the different biomass reactions available in AGORA were included as part of the core in order to guarantee biomass production in the resulting model. In the second iteration, the core comprised the resulting network from Iteration 1 and the input exchange associated with the first nutrient from i-Diet. In the third iteration, the core comprised the resulting network from Iteration 1 and the input exchange associated with the second nutrient from i-Diet. This process was repeated for the 221 nutrients from i-Diet. Reactions obtained at each iteration were included in the final model.

Note here that, in order to include each input exchange reaction as part of the core in the different iterations, we split reversible reactions in our universal database into two irreversible steps. In addition, when we added the input exchanges of nutrients from i-Diet as part of the core in the different iterations, we penalized the inclusion of their associated output exchanges to avoid artifacts in the resulting network (weight of 1e5). The same approach was employed for the output exchanges of i-Diet metabolites.

We integrated the reactions selected in the different iterations of the gap-filling process, obtaining an active network made of 2920 metabolites and 6277 reactions. At this stage, we still had 51 reactions without taxonomic assignment to species in AGORA. To avoid the inclusion of metabolic pathways with low evidence in the human gut microbiota, we deleted this subset of reactions and ran fastFVA[48], obtaining a metabolic model involving 2742 metabolites and 6122 reactions. As a result, all the reactions extracted from the universal database to fill the gaps in the degradation of input dietary compounds included taxonomic assignment to species

in AGORA, which provides a higher reliability to the outcome of the process. Note here that, for the transport reactions associated with i-Diet compounds, we assumed their presence in those species that encoded their degradation reactions and gave them the corresponding taxonomic annotation. An illustration of the gap-filling algorithm can be found in Supplementary Note 1.

We then checked mass and stoichiometry balance of reactions in AGREDA with the function checkBalance from the COBRA Toolbox[22,23]. Inconsistent reactions were manually corrected using available metabolic databases, such as KEGG or MetaCyc[40], or deleted in the case the level of accuracy or annotation in Model SEED was insufficient. Reversibility of reactions was defined according to the information provided in AGORA, Model SEED, and other metabolic databases. After corrections and removals, we extracted a metabolic model with 2720 metabolites and 6088 reactions.

Note here that each reaction in the derived metabolic model includes its associated Taxonomy-Reaction (TR) rules, which defines the OR Boolean rule with the species supporting this reaction, in analogy with Gene-Protein-Reaction rules typically used in the field of CBM. TR rules permit a straightforward contextualization of the overall model with data of species abundance as well as conducting single-species analysis. A more detailed description of TR rules can be found in Supplementary Note 1.

**Single-species analysis**. After the gap-filling process, due to the mixed-bag network strategy adopted, we may have pathways requiring transport reactions that are not defined in our metabolic model, which may lead to incorrect predictions. To overcome this issue, we conducted single-species analysis and identified for each organism metabolites involving exclusively consumption or production reactions (dead-end metabolites). The flux through this subset of metabolites is therefore blocked in their corresponding species. We identified 289 metabolites presenting this problem in at least 1 of the 818 species. Then we searched these metabolites in the Human Metabolome Database and included an exchange reaction in the following cases: (1) they were identified in both feces and biofluids in the entry "Biospecimen Locations"; (2) they were identified in feces in the entry "Biospecimen Locations" and located in the extracellular compartment in the entry "Cellular Locations." This criterion was satisfied for 37 metabolites, and therefore an exchange reaction was included in our model for each of them (Supplementary Data 3).

After including these exchange reactions, we ran fastFVA[48] for each species separately and deleted blocked reactions. Once TR rules were updated, we obtained a metabolic model, called AGREDA, that involves 2602 metabolites and 5944 reactions. AGREDA can degrade and produce 209 and 147 (out of 221) dietary compounds from i-Diet, respectively. Full details can be found in Supplementary Data 2.

**Building context-specific AGREDA and AGORA models**. From the list of metabolites and reactions involved in AGREDA, we derived the resulting stoichiometric matrix, **S**, and forced the mass balance equation:

$$\mathbf{S} \cdot \mathbf{v} = \mathbf{0} \tag{1}$$

where the **v** vector stores reaction fluxes.

Equation (2) represents lower and upper bounds (**lb** and **ub**, respectively) for reaction fluxes:

$$\mathbf{lb} \leq \mathbf{v} \leq \mathbf{ub} \tag{2}$$

Reaction fluxes are limited by irreversibility constraints, growth medium conditions (diet), and species abundances (16S rRNA gene sequencing data). Lower bounds for irreversible fluxes were made equal to zero. In addition, uptake fluxes of compounds that are not present in the diet were fixed to zero. Finally, reactions whose annotated species are not present in the sample considered were blocked, i.e., both lower and upper bounds were set to zero. For this last step, we made use of the TR rules introduced in the previous subsection.

In the analysis reported in the "Results" section, for each sample, we assessed which output metabolites can be potentially produced from input nutrients available in the different recipes. For this study, which is essentially structural, we fixed a sufficiently large bound $M$ ($M = 1000$) for the uptake fluxes of active nutrients in the different recipes.

AGREDA included a different biomass reaction for each species, which was directly extracted from AGORA. We checked that biomass production was possible for all species in AGREDA in different reported growth media. We could potentially apply to AGREDA different approaches in the literature, ranging from multi-objective approaches to models where all species grow at the same rate[49]. However, in our structural analysis, we did not force any constraint related with producing a minimum of biomass for each species in the sample. Our qualitative study is not affected by the inclusion of these constraints.

Once each context-specific model was defined, based on diet and species abundance, we applied FVA[32]. As a result, we obtained blocked reactions (their minimum and maximum fluxes are equal to zero) and potentially active reactions (their minimum or/and maximum fluxes are different from zero). As we particularly focus on output exchange reactions, we developed our own implementation of FVA in the MATLAB environment using IBM ILOG CPLEX as the optimization engine. This analysis was done for all recipes and samples considered in the "Results" section. The same methodology was applied to AGORA.

**Statistical analyses**. In the "Results" section, we discussed the relevance of different recipes and clinical conditions in the production of output microbial metabolites in the children considered. In particular, we built a logistic regression model for each output microbial metabolite using recipes and clinical conditions as (independent) explanatory variables. For each output metabolite, the binary response is a vector of dimension 84 (4 clinical conditions and 21 growth medium conditions), being 1 if the metabolite can be produced, 0 otherwise. In order to avoid multicollinearity, we took as reference (intercept) lean children under minimal growth medium conditions and kept the other 23 independent binary variables (obese, celiac, and allergic to cow's milk plus 20 recipes considered). For multiple hypothesis correction, we integrated all the $p$ values for different output metabolites analyzed and applied the false discovery rate (FDR) approach. Note here that output metabolites presenting no variation in the different cases were not considered in this analysis.

**In vitro gastrointestinal digestion and fecal fermentation of lentils**. For the in vitro digestion and fermentation, the following reagents were used: potassium di-hydrogen phosphate, potassium chloride, magnesium chloride hexahydrate, sodium chloride, calcium chloride dihydrate, sodium mono-hydrogen carbonate, ammonium carbonate, and hydrochloric acid, all obtained from Sigma-Aldrich (Germany). The enzymes—salivary alpha-amylase, porcine pepsin, and bile acids (porcine bile extract)—were purchased from Sigma-Aldrich, and porcine pancreatin was from Alfa Aesar (United Kingdom). The fermentation reagents (sodium di-hydrogen phosphate, sodium sulfide, tryptone, cysteine, and resazurin) were obtained from Sigma-Aldrich (Germany).

The in vitro digestion method was carried out according to the protocol described by Brodkorb and colleagues[50]. Briefly, in the oral phase, 5 mL of salivary solution with alpha-amylase (75 U/mL) and 25 μL of 0.3 M $CaCl_2$ were added to 5 g of lentils and the mix was incubated at 37 °C for 2 min. Then 10 mL of gastric solution with pepsin (2000 U/mL) and 5 μL of 0.3 M $CaCl_2$ were added and the pH was lowered to 3.0 by adding 1 N HCl; the mix was then incubated at 37 °C for 2 h. Finally, 20 mL of intestinal solution with pancreatin (100 U/mL), bile salts (10 mM), and 40 μL of 0.3 M $CaCl_2$ were added and the pH was raised to 7.0 with 1 N NaOH, after which the mix was incubated at 37 °C for 2 h. The enzymatic reactions were halted by immersing the tubes in iced water. The samples were then centrifuged at $6800 \times g$ for 10 min at 4 °C and the supernatants were separated from the solid residue or pellet.

The in vitro fermentation was carried out according to the protocol described by Pérez-Burillo et al.[51,52]. Feces were collected from three children (9–11 years old) from each of the groups studied: allergic to cow's milk, celiac, obese (body mass index (BMI) ≥ 30) and lean (BMI ≤ 25). Feces from children belonging to the same group were pooled together to account for inter-individual variability. Additionally, seven different inocula were prepared from the celiac-, lean-, and obese-derived pools, respectively, and six different inocula from the allergic-derived one, yielding therefore a total of 27 fermentation experiments. Right after collection, feces were mixed with glycerol (50:50 w/v) and frozen at −80 °C. Briefly, 500 mg of digested wet-solid residue were placed in a screw-cap tube. In all, 10% of the digestion supernatant was added to the solid residue in order to mimic the fraction that is not readily absorbed after digestion. Then 7.5 mL of fermentation medium (15 g/L of peptone, 0.312 mg/L of cysteine, and 0.312 mg/L of $Na_2S$, adjusted to pH 7.0) and 2 mL of inoculum (consisting of a solution of 32% feces in phosphate buffer 100 mM, pH 7.0) were added, to reach a final volume of 10 mL + digestion supernatant volume. Nitrogen was bubbled through the mix to produce an anaerobic atmosphere and the mix was then incubated at 37 °C for 20 h under oscillation. Immediately afterwards, the samples were immersed in ice, to stop microbial activity, and centrifuged at $6800 \times g$ for 10 min. The supernatant, representing the soluble fraction potentially absorbed after fermentation, was collected and stored at −80 °C.

**DNA extraction and amplicon sequencing**. Genomic DNA from the solid residues of the fermentation reactions was extracted using the MagNaPure LC JE379 platform (ROCHE) and the DNA Isolation Kit III (Bacteria, Fungi) Ref. 03264785001, following the manufacturer's instructions, with a previous lysozyme lysis. DNA was quality-checked by agarose gel electrophoresis (0.8% wt/vol agarose in Tris-acetate-EDTA buffer) and quantified using the Qubit 3.0 Fluorometer (Invitrogen) and the Qubit dsDNA HS Assay Kit.

In order to prepare amplicon libraries, DNA at 5 ng/μL in Tris 10 mM (pH 8.5) was used for the Illumina protocol for the small subunit ribosomal RNA gene (16S rRNA) Metagenomic Sequencing Library Preparation (Cod 15044223 Rev. A). PCR primers targeting the V3–V4 hypervariable region of the 16S rRNA gene were designed as described by Klindworth et al.[53] (Supplementary Table 6). These primers contain adapter sequences added to the gene-specific sequences to make them compatible with the Illumina Nextera XT Index Kit (FC-131-1096). After 16S rRNA gene amplification and indexing, amplicons were multiplexed and sequenced in an Illumina MiSeq sequencer according to the manufacturer's instructions in a 2 × 300 cycle paired-end run (MiSeq Reagent Kit v3MS-102-3001).

**Taxonomic assignment of 16S rRNA gene sequencing data.** 16S rRNA gene raw sequence reads were processed, trimmed, and clustered into amplicon sequence variants (ASVs) using DADA2[54]. Once we obtained the ASV table, we assigned species-level taxonomic identifications to each ASV with DADA2, based on exact matching (100% identity) between ASVs and the reference sequences in the Silva database (version 132)[55].

In addition, for those ASVs that were identified with DADA2 at the genus level but not at the species level, we applied the MegaBLAST module from BLAST[56]. Here we required at least 97% identity for the species-level assignment; however, as MegaBLAST does not take into account the previously assigned genus level, we only considered ASVs for which MegaBLAST and the DADA2 classifier method assigned the same genus. Finally, ASVs with <0.01% of the total number of counts were removed, and rarefaction was applied up to the smallest library size across samples (52,923 counts) for further analysis.

Finally, we linked identified taxa to the species present in AGORA. Due to the metabolic diversity between organisms within each genus, we only considered ASVs that reached the species level. Given that the taxonomic assignment methods typically provide information at the species level but not at the strain level, identified taxa could be related to different strains present in AGORA. To overcome this issue when contextualizing data in AGREDA, we considered reactions common for all the strains within the species. We followed this strict strategy to avoid including reactions with limited taxonomic evidence.

**Identification and quantification of phenolic compounds.** For individual phenolic quantification, the following standards were used: phloroglucinol, phenol, 3-(3-hydroxy-phenyl)propionate, 4-hydroxyphenylacetate, (3,4-dihydroxyphenyl) acetate, dihydrocaffeic acid, cianidanol, myricetin, and quercetin (all purchased from Sigma-Aldrich, Germany) and 5-(3′,4′-dihydroxyphenyl)-gamma-valerolactone (purchased from Toronto Research Chemicals, Canada). Diethyl ether for extraction was purchased from Sigma-Aldrich (Germany).

Phenolic compounds were analyzed through ultraviolet ultra-high-performance liquid chromatography (UHPLC) as described by Pérez-Burillo et al.[57], slightly modified to adapt it to UHPLC. In brief, 1 mL of fermentation supernatant was mixed with 1 mL of diethyl ether and kept in the dark at 4 °C for 24 h. The organic phase was then collected and two more extractions with diethyl ether were performed. These 3 mL of diethyl ether were dried in a rotary evaporator set at 30 °C, and the solid residue was resuspended in 1 mL of methanol:water (50:50 v/v) mix. The mixture was then ready to be injected into the UHPLC system (Agilent 1290 Infinity II), which is equipped with a quaternary pump, an autosampler kept at 5 °C, and a diode array detector set at 255 nm. The column used was an InfinityLab Poroshell 120 Sb-Aq 2.1 × 150 mm and 1.9 microns. The flow rate was set at 0.250 mL/min for 46 min. Two mobile phases were used: milli-Q water with 0.1% of formic acid (A) and acetonitrile (B) with the following gradient: 0 to 28 min from 95 to 60% of A and from 5 to 40% of B; 28–36 min from 60 to 0% of A and from 40 to 100% of B; 36–41 min from 0 to 95% of A and from 100 to 5% of B; these last conditions were kept for 5 min. Identification and quantification were carried out by comparing retention times obtained from pure standards (listed in reagents section). A calibration curve for each of the compounds was performed in the range of 0.1–25 p.p.m.

**Untargeted metabolomics.** The untargeted analysis of the metabolites produced after in vitro fermentation of lentils was performed following the method of Rocchetti et al.[58]. Briefly, fermented extracts were filtered through nylon filters of 0.20 μm before UPLC injection (2.5 μL). A quality-control sample was prepared by mixing equal volumes from each fermented extract. The quality-control sample was randomly injected during the sample analysis to monitor the system stability and attenuate the analytical variation resulting from system instability.

A LC apparatus ACQUITY UPLC M-Class System (Waters Corp., Milford, MA, USA) was coupled to a time of flight–mass spectrometer detector (SYNAP G2 from Waters). The mass spectrometer and UPLC system were controlled by the MassLynx v4.1 software. The UPLC column was a fused-core Poroshell 120, SB-C18 (3.0 × 100 mm, 2.7 μm; Agilent Technologies, Palo Alto, CA, USA). The mass spectrometer was operated in both negative and positive modes. A gradient elution was programmed using as a mobile phase A acidified water (1% acetic acid) and as mobile phase B acetonitrile. A linear gradient was applied as described in Gómez-Caravaca et al.[59]: 0 min, 5% B; 12.5 min, 20% B; 17.5 min, 60% B; 22 min, 5% B. The initial conditions were maintained for 5 min. The flow rate was set at 0.6 mL/min throughout the gradient. Separation was carried out at 25 °C. The mass spectrometric (MS) analyses were carried out in full-scan mode (range m/z 50–1000) using an electrospray (electrospray ionization) interface and the following conditions: drying gas flow, 9.0 L/min; nebulizer pressure, 35 psig; gas drying temperature, 350 °C; capillary voltage, 3000 V; and fragmentor voltage, 80 V.2.5. All MS data were acquired using the LockSpray to ensure reproducibility and mass accuracy. The molecular masses of the precursor ion and of product ions were accurately determined with leucine enkephalin (m/z 556.2771) in negative and leucine enkephalin (m/z 554.2615) in positive modes at the concentration of 1 ng/μL at an infusion flow rate of 5 μL/min.

For data analysis, features were selected based on their coefficient of variation with the quality-control sample (features with a coefficient of variation >30% were eliminated). Raw data were processed using the MassLynx v4.1 software (Waters, USA) according to the targeted "find-by-formula" algorithm. Accurate mass information was used together with the spectral isotope pattern (isotopic spacing and isotopic ratio) to achieve a higher confidence in metabolite identification. Data from Human Metabolome Database and Phenol-Explorer 3.6 were used as references for compound identification, adopting a 5 p.p.m. tolerance for mass accuracy. Thus, identification was carried out according to Level 2 (i.e., putatively annotated compounds) as set out by the COSMOS Metabolomics Standards Initiative (http://cosmos-fp7.eu/msi). Potential metabolites that passed the mass accuracy detection threshold, that had plausible chromatogram peak features, and that showed significantly different trends from the control (fecal fermentation without lentils) were considered as potential fermentation markers of the different conditions.

**Ethics statement.** Informed consent was obtained from all participants in accordance with the Declaration of Helsinki. This study was approved by the Ethics Committee of the University of Granada (protocol code 1080/CEIH/2020, approved 10/06/2020).

**Reporting summary.** Further information on research design is available in the Nature Research Reporting Summary linked to this article.

## Data availability
The 16S rRNA sequencing data were obtained within the STANCE4HEALTH research project and are available at https://www.ebi.ac.uk/ena/browser/home under accession code PRJEB40603. The metabolomics data are provided in Supplementary Data 3. The rest of the data employed in this study can be obtained from the following databases: (i) AGORA metabolic models: Virtual Metabolic Human (https://www.vmh.life/); (ii) Universal metabolic databases: The Model SEED (https://modelseed.org/), Kyoto Encyclopedia of Genes and Genomes (https://www.genome.jp/kegg/); (iii) Genome FASTA files: GenBank (https://www.ncbi.nlm.nih.gov/genbank/), Ensembl (https://www.ensembl.org/index.html); (iv) Taxonomic annotation of metabolic capabilities: MetaCyc (https://metacyc.org/), Brenda (https://www.brenda-enzymes.org/), UniProt (https://www.uniprot.org/); (v) Metabolites: PubChem (https://pubchem.ncbi.nlm.nih.gov/), Human Metabolome DataBase (https://hmdb.ca/), RetroRules (https://retrorules.org/), i-Diet (http://www.i-diet.es/), and Phenol-Explorer (http://phenol-explorer.eu/).

## Code availability
The source code to generate AGREDA can be found in https://github.com/tblasco/AGREDA.

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

## Acknowledgements
This work was funded by the European Union's Horizon 2020 research and innovation program through the STANCE4HEALTH project (Grant No. 816303).

## Author contributions
M.P.F, J.A.R.-H., and F.J.P. conceived this study. T.B., F.B., X.C., A.R., I.A., and F.J.P. developed the metabolic network and performed the computational analysis. S.P.-B, D.H.-N, S.P., and J.A.R.-H. carried out the in vitro fermentations, measured the phenolic compounds, and conducted untargeted metabolomics. M.J.G, A.L.-A., N.J.-H., and M.P.F performed the metagenomics analysis. All authors wrote, read, and approved the manuscript.

## Competing interests
The authors declare no competing interests.
