## [Peer Review File · Nature Communications]

Reviewers' Comments:

Reviewer #1:

Remarks to the Author:

The authors' aim was to extend the existing analysis AGORA to include a larger number of metabolites, allowing for more accurate analysis of microbial metabolism of dietary constituents. The authors describe the successful addition of 231 nutrients and validate their new analysis by applying it to an in vitro fermentation experiment, noticing superior performance of AGREDA compared to its processor. As such, AGREDA could provide further insight into microbial metabolism of dietary components and as such lends itself to be of significance to the field, especially in the effort to understand how microbial metabolites could be involved in microbial signaling and influencing host processes. Therefore, the manuscript could have implications for a variety of researchers, including nutrition researchers as well as researchers who are investigating underlying mechanisms of dietary modulation of the microbiome and the impact on host processes, by providing more accurate insight in predicting microbial metabolites of dietary compounds.

Mostly the claims in the manuscript are convincing. However, it appears that the majority of the added metabolites are from polyphenol degradation. The authors should make this clear from the beginning of the manuscript that the focus of this extended analysis is on polyphenol metabolism (such as is mentioned in line 262). There are a lot of other nutrients and metabolites that are important in personalized nutrition approaches, so it is misleading to call it "improvement of the effect of diet on microbial metabolism" (line 273-274) when really the only improvement was made in one nutrient class. Thus, the claim made by the authors that the new analysis provided by AGREDA will "improve the predictive capacity of gut microbial metabolism" (line 193-196) is not convincing.

Additionally, the discussion could be improved, for example, by focusing more on the translatability of the results. In the current form, the authors seem to mostly summarize the results and putting it in little context with the existing literature. In the introduction, the authors claim that this approach could drive the "personalized nutrition approach", but it is not clear how exactly they envision this. Can the new information provided be used to characterize baseline microbial composition/function of patients which can then be translated into nutrition approaches? Or will clinicians be able to tell patients how much of which polyphenols they should eat? In the current state, the manuscript is not convincing regarding the translatability to personalized nutrition approaches, as claimed in the title. It appears that this approach might be more useful in post analyses rather than predicting metabolic capacity of the microbiome to develop targeted nutrition approaches.

Another concern that is arising is the small amount of representative phenol compounds investigated to validate that AGREDA is superior to AGORA. Out of the 231 metabolites that were added, the authors only investigated 10 in their experiment. This is a very small percentage and poses the question if it truly is a validation experiment to confirm the prediction power of AGREDA. Might an untreated HPLC approach be more useful in determining if indeed AGREDA is indeed superior to AGORA?

Overall, the manuscript is written in clear English and understandable. Only few minor things should be addressed:

Line 159-160 the sentence "We observed that the latter is significantly greater than the former" is not needed.

Also the authors describe the group of children with cow's milk allergy as "allergic" or having "allergy to food". The specific allergy should be clarified throughout the manuscript.

There is repetition in some places, for example, Lines 297-299 and 349-350 regarding "phenol compounds being important for personalized nutrition". Please carefully revise the manuscript concerning repetitions.

I do not have enough expertise in bioinformatics to judge the methodological technicalities of the

manuscript. However, it appears that sufficient detail has been provided. The only thing I would question is the authors mentioning of "expert opinion" without ever describing what exactly they are referring to.

Reviewer #2:

Remarks to the Author:

Blasco et al expanded the well-known AGORA model by data-driven curation and performed a case study to predict the effect of lentil diet on gut microbiome in different conditions. Here, the authors did a lot of annotation and manual work, however AGREDA seems to be a minor extension to AGORA with added degradation pathways of selected nutrients. Other concerns are listed below:

1. Supra organism strategy should be explained clearly. Connection of individual models to each other by keeping taxonomical assignation seems just removing duplicating reactions in AGORA models. How does this model predict the complex microbial interactions?

2. Which AGORA version is used? How did the authors define the non-redundant set of metabolites?

3. How can a researcher use this model? For example, if one wants to create a bacteria model by using AGREDA she should keep related rows? What will be the biomass reaction for bacteria of interest? Is the model stoichiometrically balanced for each bacterium in the model?

4. There is unnecessarily detailed information about annotation of SEED and other databases. This can be briefly written and move to a detailed supplementary appendix. However, the most important part is creating the AGREDA model and its usage should be written clearly.

5. How is the objective function of AGREDA generated? What is the biomass reaction for FVA?

6. Is AGREDA model mass- and energy-balanced?

7. In the core of the used gap filling algorithm, what are the rationale of the values for weights were defined? Were all the included reactions obtained after each iteration compatible with each other, if not how the reactions were chosen?

8. This work is highly focused on polyphenols and case study is conducted on a polyphenol rich food. A normal diet constitutes numerous bioactive molecules. Is the proposed solution biologically relevant?

9. The phenolic compounds added to the model. They are already mentioned in the literature as allergenic components. In other words, the model becomes an allergic-specific model in this respect. Therefore, it is normal for allergic pathways or metabolism to change much more. And it may not be very suitable for lean, obese, and celiac situations.

10. Several times, AGORA was defined as the largest repository of gut bacterial metabolic reconstructions in the literature. However, Machado et al (reference no 8) automatically created 5587 bacterial models by using the CarveMe tool.

11. It is stated that AGREDA is more accurate and sensitive, however, it is not clear what the comparison was made. I suggest the authors perform comparative analysis with AGORA and other models available. It is very well known that AGORA models have major issues which have been discussed by Nielsen's group in a commentary at Nature Biotechnology.

12. The model is separated as external and internal, and the borders of each species have been removed. The necessary reactions were selected from the reaction pool. In this case, are the reactions

representing each species balanced in themselves? This is important and not mentioned in the text.

13. The number of samples for each case is small. The authors should discuss this in the paper carefully.

14. Line 176 - they should change the broad term "allergic to foods" to "cow's milk allergic" like they specify in the Methods.

15. Line 189 - Strictly speaking the AGORA network predicts 5 out of 10 measured phenolic compounds with true negatives detected for quercetin and 34dhpgval in the lean subjects.

16. Line 285 - It would be ideal if this further experimental validation in a larger cohort of children could be performed and included as part of this paper. Three samples in each clinical group is a very low number.

Reviewer #3:

Remarks to the Author:

This paper is well written and addressed important knowledge gaps in the metabolic pathways of phytochemical conversion in the gut microbiota. However, I am afraid that this paper seems to fail to demonstrate the fidelity and performance of the model (AGREDA) and its genuine connection to personalized diet/nutrition. In addition, I appreciate the use of the authors' own in vitro experimental data in this paper, but I am afraid that the specific methods used for the model validation based on these experimental data do not sound rigorous. The followings are my detailed comments.

1. The claim of the connection of this work with "personalized diet/nutrition" seems to be a bit over-selling. For example, did this paper recommend any diet or nutrition specific to each donor of the fecal samples in the paper? In my view, the modeling results in this paper are just the identification of sample-specific metabolic pathways, without much specific dietary implications. Therefore, I am afraid that the connection of this paper to personalized diet/nutrition deserves to be mentioned in the Discussion section to a smaller degree, rather than over-emphasized as in the current manuscript like the main theme of the paper. Another issue is that this paper seems to give an impression that their contextualized network of individual fecal samples are disease-specific. However, the number of these samples for each disease group is not large enough to control for any confounding factors in the analysis of the disease group-associated data, and also this paper didn't try to control for these confounding factors. Therefore, I am not convinced about whether their analysis results are really disease-specific or are the reflection of other features of the sample donors.

2. The authors justify the use of the supra-organism-like pathways as for the saving of computational costs. However, the supra-organism approach can radically over-simplify the gut community metabolism driven by inter-species metabolic interactions in the gut. Hence, rigorous theoretical justification and detailed assessment of their approach should be provided. Otherwise, even the model outcome is backed-up by some experimental data, this can be potentially "lucky" match between the model results and the presented particular experimental data, rather than indicates robust consistency with experimental data in various settings.

3. This paper doesn't seem to compute any fluxes in the metabolic pathways using FBA. For example, during the model construction, biomass reactions of the supra-organism network do not seem to be designated for the use in the gap-filling processes. If this gap-filling process was indeed done without any biomass reactions, the gaps would have been filled anyhow without a need for microbial growth. This can be a source of substantial false-positive or false-negative reactions after the gap-filling, because the gap-filling results are likely to be highly-dependent on the specific form of the biomass reactions.

Because the authors did not seem to solve FBA (with specific biomass reactions) when running their constructed model either, the model results are just likely the "network topology"-based ones. Such a pure topology analysis does not really "predict" activated metabolic pathways for a given set of input nutrients. To infer such activated metabolic pathways using the network topology, the contextualization of the network needs to be done (as in this paper) with the substantial amount of empirical data e.g. 16S rRNA gene sequencing data and other omics data. To me, relying on such substantial experimental data itself questions whether this method is indeed "predictive" enough for unknown facts. Moreover, in this case of the contextualization of the supra-organism network topology, I wonder what would be the fundamental difference between the approach in this paper and the previously-established method of the network contextualization such as HUMANN. Together, Without de novo flux calculation, the use of genome-scale metabolic models is limited to the topology analysis, and hence the full power of the genome-scale metabolic model on the prediction of the flux values was not harnessed in the paper. In other words, the authors' claim on the true value of genome-scale metabolic models (on the personalized nutrition) is not supported enough in the paper.

4. Transporters in known genome annotations are notoriously poorly annotated, even for primary metabolites. For the plant secondary metabolites focused in this paper, the annotation quality might be even worse. On the other hand, the presence/absence of transporters in an organism is not even always consistent with the true nutrient transport in the microbe, because a number of metabolites can freely diffuse into/out of the cell without the help of the transporters. In the case of phytochemicals focused in this paper, they may also include membrane-diffusible compounds (such as fat-soluble metabolites). These parts are not adequately considered in ModelSEED that the authors used. Actually, it can be potential sources of numerous errors in the model AGREDA in the paper. In addition, I am afraid of other model quality issues as well, including possibly many false-positive reactions, as discussed below.

In Line 161, "We can, therefore, conclude that AGREDA provides us with a more accurate tool to assess the effects of the different diets on the gut metabolism with a straightforward application to personalized nutrition." I am afraid that this sentence sounds rather overselling. The main reason of this analysis result (regarding the compounds in different recipes) is that AGREDA includes more pathways for the degradation of many phytochemicals, so AGREDA naturally has more power to differentiate different recipes based on their compositions. However, it does not automatically guarantee that AGREDA is "more accurate tool" (the phrase used above) than AGORA. Inclusion of more pathways may mean the presence of more false-positive pathways, as well. In addition, I wonder how this result can be "with a straightforward application to personalized nutrition" in the above sentence.

Line 189: "the reference (uncontextualized) AGORA network only captures 3 out of 10 measured phenolic compounds, while the reference (uncontextualized) AGREDA network contains all the measured metabolites." Although AGREDA may have better sensitivity than AGORA due to more compounds/pathways added to AGREDA, there would be higher risk of false positives in AGREDA, as well. This point needs to be addressed. If Line 193 ("AGREDA outperforms AGORA regarding accuracy (75% versus 32,5%)") intended to address this issue of false positives, the definition of the accuracy here should be clearer. The accuracy should be varied across different samples, but it is not clear what these aggregate numbers (75%, 32.5%) mean.

The authors may claim that false-positive issues have already been assessed through their own metabolomic data, like "FP" in Figure 4. However, Figure 4 is only for several enriched metabolites in each sample. and many false-positive results are likely to be present in the list of "all" metabolites predicted by the AGREDA for each sample. It'd better compare the comprehensive list of these predicted metabolites and that of the detected metabolites in the metabolome data for each sample and identify the false positive rates (rather than focusing only on a smaller set of the enriched metabolites).

5. In Line 194, "We, therefore, conclude that the new metabolites and degradation pathways included in AGREDA significantly improve our predictive capacity of gut microbiota metabolism and enable the detection of output metabolites not considered in AGORA." To prove the better "predictive"

power of AGREDA than that of AGORA, it is important to validate "output" of the models, as also written in the above sentence. However, it is not clear whether the accuracies have been only calculated for intermediate-to-terminal metabolites in the pathways, rather than the inclusion of input phenolic compounds of which presence may simply dominate the overall accuracy values.

6. Most of "nutrients" newly added to this model seem plant secondary metabolites. However, these metabolites are not usually called "nutrients", but rather the use of the word phytochemicals is more accurate and clearer. They are classified to neither macronutrient nor micronutrient.

7. Many of plant secondary metabolites are known to have anti-microbial effects (the authors seem already aware of this fact as Line 137 includes a phrase "they act as a defensive system"). If these metabolites in the paper have such anti-microbial effects (which may dominate their effects as consumable metabolites of the microbes in the paper), then how does the validity of the paper's claim on the efficacy of their models be justified? For example, benzoic acid in Line 140 is used as one of the famous chemical food "preservatives" to suppress microbial growth, to my knowledge.

8. The word "nutrient degradation" is not clearly defined in the introduction or in the front of the Results in the main text. Some audience may initially wonder whether "nutrient degradation" means the pathway of macromolecular degradation to small molecules (that usually happen outside cell membrane) or the entire biotransformation pathway of nutrients to their terminal downstream metabolites (that involves metabolic reactions inside cells). Even after some audience realize that the nutrients in this paper mainly mean phytochemicals, some of them may still wonder whether the degradation pathways considered in this paper cover the literally only the "breakdown" pathways of these phytochemicals, or the entire biochemical reactions regarding these phytochemicals.

The followings are relatively minor points:

Line 181: 16S rRNA gene sequencing is able to detect usually genus-level taxa. It should be clarified how the authors mapped these identified genera to strain-level AGORA models considered in AGREDA.

Line 49: "Systems Biology" => "systems biology"

Line 61: "For example, AGORA has been already applied to predict dietary supplements for Crohn's disease [15]. Using a similar approach, we predicted the effect of solid diet on the gut microbiota metabolism of infants [16]."

==> The efficacy of the existing genome-scale metabolic models in the design of personalized diets should not be over-emphasized. In the case of Ref. [15], its used genome-scale metabolic models (AGORA) are not yet in the form of realistic platform for the computational simulation of the gut environment. For example, AGORA lacks the species-specific growth rates (e.g. without ATP maintenance energy adjustment) and its simulation platform does not consider characteristic environmental factors in the large intestine such as the nutrient mixing by peristalsis that is known as important for gut microbial dynamics. As demonstrated in Ref. [15] itself, the simulated time period is also too short to realize a biologically meaningful gut microbial dynamics. And the "predicted" diet supplements in Ref. [15] were not experimentally validated. In the case of Ref. [16], the claimed experimental validation is rather limited to a particular metabolite and is viewed as rather weak to guarantee the promising value of the existing genome-scale models or networks in the context of personalized diet/nutrition design.

Line 80, 180: "16S rRNA sequencing", "16S sequencing"

==> "16S rRNA gene sequencing". Check the appropriate word usage from Microbiome 3, 31 (2015).

Line 86: "genomics"

==> Shouldn't this be "metagenomics" ?

Line 144: "The daily intake of phenolic compounds is rather high"

==> I am not sure whether this sentence can be viewed as correct, because of the amount of plant secondary metabolites is very low in a food compared with macro- or micronutrient contents, although they are contained in many plant-based foods.

Line 160: "AGREDA performs better at capturing the potential metabolic differences between the recipes."

==> The word choice of "potential metabolic differences" sounds like over-selling, because this analysis was simply the analysis of the compounds in the recipes overlapped with the metabolites in AGORA and AGREDA. The phrase "potential metabolic differences" sounds like something about the simulated model outcomes under these two recipes.

Line 268, 270: "Phenol-Explorer" and "SEED" are written without citation here, but it should be cited.

LINE 199: "<=" should be written as math symbol for 'less than or equal to'. And, it is not clear whether the authors performed FDR correction of the P values.

Line 377: "To avoid false positives, we deleted this subset of reactions"

==> The specific criteria for deleting these reactions should be clearly described in detail.

Reviewer #1 (Expertise: Host-microbiome interactions):

The authors' aim was to extend the existing analysis AGORA to include a larger number of metabolites, allowing for more accurate analysis of microbial metabolism of dietary constituents. The authors describe the successful addition of 231 nutrients and validate their new analysis by applying it to an in vitro fermentation experiment, noticing superior performance of AGREDA compared to its processor. As such, AGREDA could provide further insight into microbial metabolism of dietary components and as such lends itself to be of significance to the field, especially in the effort to understand how microbial metabolites could be involved in microbial signaling and influencing host processes. Therefore, the manuscript could have implications for a variety of researchers, including nutrition researchers as well as researchers who are investigating underlying mechanisms of dietary modulation of the microbiome and the impact on host processes, by providing more accurate insight in predicting microbial metabolites of dietary compounds.

Response: We appreciate the positive comments made by the reviewer.

Mostly the claims in the manuscript are convincing. However, it appears that the majority of the added metabolites are from polyphenol degradation. The authors should make this clear from the beginning of the manuscript that the focus of this extended analysis is on polyphenol metabolism (such as is mentioned in line 262). There are a lot of other nutrients and metabolites that are important in personalized nutrition approaches, so it is misleading to call it "improvement of the effect of diet on microbial metabolism" (line 273-274) when really the only improvement was made in one nutrient class. Thus, the claim made by the authors that the new analysis provided by AGREDA will "improve the predictive capacity of gut microbial metabolism" (line 193-196) is not convincing.

Response: We were not sufficiently clear in the previous version of the manuscript. In the new version of the manuscript, we specify that AGREDA includes 217 input dietary compounds, from which 183 are phenolic compounds, which are poorly described in AGORA. The remainder 34 compounds fall in different families. Particularly, we improved carbohydrates, carotenoids, fats, minerals, phytosterols, vitamins and xanthines. This is now detailed in Figure 2d and associated text:

"In addition to phenolic compounds, we improved the coverage of other relevant families of metabolites present in the diet, ranging from carbohydrates, carotenoids, fats, minerals, phytosterols, vitamins and xanthines (Figure 2d). Overall, AGREDA included 34 dietary compounds from these families that are not present in AGORA."

Additionally, the discussion could be improved, for example, by focusing more on the translatability of the results. In the current form, the authors seem to mostly summarize the results and putting it in little context with the existing literature. In the introduction, the authors claim that this approach could drive the "personalized nutrition approach", but it is not clear how exactly they envision this. Can the new information provided be used to characterize baseline microbial composition/function of patients which can then be translated into nutrition approaches? Or will clinicians be able to tell patients how much of which polyphenols they should eat? In the current state, the manuscript is not convincing regarding the translatability to personalized nutrition approaches, as claimed in the title. It appears that this approach might be more useful in post analyses rather than predicting metabolic capacity of the microbiome to develop targeted nutrition approaches.

Response: We are very grateful for the reviewer's comment. We have added a new section to the manuscript aiming to address this question: "In-silico prediction of metabolic interaction

between diet and children's microbiota". In addition, we have modified the Discussion section to envision how AGREDA can help in the development of personalized nutrition programs.

In summary, for the same Spanish children considered in the study of lentil fermentation, AGREDA was used to predict the list of output microbial metabolites that can be potentially obtained from 20 different typical recipes. We could identify output microbial metabolites whose production is linked to the gut microbiota of specific clinical conditions, based on a limited number of samples, but also to specific recipes or to both factors. We obtained a higher number of significant metabolic associations between diet and gut microbiota than AGORA, which illustrates that AGREDA is able to capture diet-specific metabolic pathways that are not present in AGORA. This is important for developing personalized nutrition programs, since the regulation of key output microbial metabolites through diet is an essential part of it. These ideas have been explicitly mentioned in the text.

Finally, we have also changed the title of the manuscript by a new one that better reflects the work presented, namely "An extended reconstruction of human gut microbiota metabolism of dietary compounds".

Another concern that is arising is the small amount of representative phenol compounds investigated to validate that AGREDA is superior to AGORA. Out of the 231 metabolites that were added, the authors only investigated 10 in their experiment. This is a very small percentage and poses the question if it truly is a validation experiment to confirm the prediction power of AGREDA. Might an untreated HPLC approach be more useful in determining if indeed AGREDA is indeed superior to AGORA?

Response: In AGREDA we included 217 input dietary compounds that could be found in many different types of plant foods after microbial fermentation, such as cereals, fruits, vegetables, etc. That means that not all the compounds included in AGREDA are found in every food. In the case of lentils, we found 63 out of these 217 compounds.

However, our interest here is to predict output microbial metabolites that can be derived from the fermentation of lentils. In other words, input dietary compounds found in lentils are prior information that is given to our metabolic model in order to predict output microbial metabolites. This section has been fully re-written to clarify the analysis conducted. We summarize below the most important points:

- On average for each sample, AGREDA identified 90 output microbial compounds that were not found in AGORA. They come from diet-specific metabolic pathways included in AGREDA but not in AGORA. For the rest of output metabolites, we obtained the same results.
- In our experimental validation, we assessed 10 output phenolic compounds in 12 different samples (120 cases). Statistical comparison is now shown in Figure 4, where AGREDA clearly overperforms AGORA.
- We found that 6 metabolites out of the 10 metabolites selected were predicted as active in AGREDA but not in AGORA at least once. In these cases, we found 3 false positives out of 60 positive predictions by AGREDA. This is a highly significant result (Hypergeometric test p-value: $2.45e-14$), and, in our opinion, it provides evidence to support the rest of predictions made by AGREDA.

Clearly, an untargeted HPLC-MS approach would be more informative and complete. However, in the light of statistical analyses reported in the manuscript, we believe that the results here presented are sufficient to validate AGREDA.

Overall, the manuscript is written in clear English and understandable. Only few minor things should be addressed:

Line 159-160 the sentence "We observed that the latter is significantly greater than the former" is not needed.

Response: This sentence was deleted.

Also the authors describe the group of children with cow's milk allergy as "allergic" or having "allergy's to food". The specific allergy should be clarified throughout the manuscript.

Response: We have modified the whole manuscript and used 'allergic to cow's milk'.

There is repetition in some places, for example, Lines 297-299 and 349-350 regarding "phenol compounds being important for personalized nutrition". Please carefully revise the manuscript concerning repetitions.

Response: We have gone through the text to avoid repetitions. In particular, we have revised the sentences mentioned by the reviewer.

Minor comments

I do not have enough expertise in bioinformatics to judge the methodological technicalities of the manuscript. However, it appears that sufficient detail has been provided. The only thing I would question is the authors mentioning of "expert opinion" without ever describing what exactly they are referring to.

Response: The comment of the reviewer is pertinent. In fact, we meant manually curated information extracted from literature. References as to reactions and metabolites extracted from literature can be now found in Supplementary Data 2. The expression "expert opinion" was substituted by manually curated literature knowledge.

Reviewer #2 (Expertise: Metabolic modeling):

Blasco et al expanded the well-known AGORA model by data-driven curation and performed a case study to predict the effect of lentil diet on gut microbiome in different conditions. Here, the authors did a lot of annotation and manual work, however AGREDA seems to be a minor extension to AGORA with added degradation pathways of selected nutrients.

Response: We would like to thank the reviewer for the comments, which helped us much to improve the manuscript.

However, we strongly disagree in that AGREDA seems a minor extension of AGORA. AGREDA substantially expands the intersection between nutrition and microbial models, including the degradation pathways of 217 nutrients in the diet that are known to be metabolized by the human gut microbiota, but are not included in AGORA. This is a major contribution since AGORA only includes 99 out of 650 nutrients detailed in the nutritional software (i-Diet) that we are using in the framework of STANCE4HEALTH (<https://www.stance4health.com/>), a European Commission research project about personalized nutrition from which this work is derived.

In the new version of the manuscript, we have made clear that AGREDA not only provides a better connection to diet than AGORA, but it also predicts diet-specific output metabolites that are not found in AGORA, providing a more complete view of the interaction between diet and gut microbiota.

Other concerns are listed below:

1. Supra organism strategy should be explained clearly. Connection of individual models to each other by keeping taxonomical assignation seems just removing duplicating reactions in AGORA models. How does this model predict the complex microbial interactions?

Response: The Methods section of the new version of the manuscript contains a more exhaustive description of the supra-organism strategy, which is now called mixed-bag network strategy, termed introduced in Henry *et al.* (2016), where a detailed assessment and justification of this approach is provided. In addition, a toy example has been included in the Supplementary Note 1 to clarify this point.

In brief, first, all the interactions between the species are modeled via the extracellular metabolites they share. Next, only one reaction is kept per duplicated reaction, but the taxonomic assignation is maintained. In other words, in the final model, reactions are not duplicated but the information on which species are in charge of performing each metabolic transformation is known. This information is stored in the Taxonomy-Reaction (TR) rules in the same way as it is typically done with Gene-Protein-Reaction (GPR) rules in genome-scale metabolic networks. Therefore, a reaction will be potentially active in those situations where at least one of the associated species is present.

Note here that, as the interactions between species are considered in the first step, they will indeed be included in our resulting mixed-bag network model. Although it is not developed in this manuscript, different constraint-based models could be easily applied to recover complex interactions between species. In a different work, we are meant to apply the concept of minimal cut sets to extract this type of interactions in mixed-bag network models.

2. Which AGORA version is used? How did the authors define the non-redundant set of metabolites?

Response: We have employed AGORA 1.03 (with mucins), the latest version published by Thiele and colleagues at their webpage. This has been mentioned in the text.

Regarding the issue with the metabolites, this was not correctly described in our previous version of the manuscript. Once the boundaries between different species are taken away, we only delete duplicated reactions and not metabolites. This sentence has been rephrased.

3. How can a researcher use this model? For example, if one wants to create a bacteria model by using AGREDA she should keep related rows? What will be the biomass reaction for bacteria of interest? Is the model stoichiometrically balanced for each bacterium in the model?

Response: The inclusion of the Taxonomy-Reaction (TR) rules in the model permits a straightforward contextualization of the overall model to a situation where only a subset of bacterial species is present. As noted above, analogously to what is done to integrate -omics data with Genome-Scale Metabolic Networks via GPR rules, TR rules permit the creation of models where not all the species take part and, therefore, enable the integration of 16S rRNA gene sequencing data with AGREDA. Briefly, in the contextualization process, only those reactions that are annotated to at least one present species will remain in the final context-specific model. TR rules have been described in the Methods section and in Supplementary Note 1.

With regards to the biomass reactions of the different models, due to their uniqueness –each species contains its own biomass reaction–, they are all included in the final model and annotated to the respective species. Therefore, we could potentially apply to AGREDA different approaches in the literature to handle this situation, ranging from multi-objective approaches to models where all species grow at the same rate (Magnúsdóttir and Thiele, 2018).

On the other hand, AGREDA is stoichiometrically balanced, *i.e.* all the reactions, including the different biomass reactions, can perform in steady-state when they are modeled as a mixed-bag network. However, if we build the metabolic network of a specific bacterium, based on AGREDA, some of the reactions could be blocked. We recognize that this is something we need to address in future versions of AGREDA, particularly in the case where we are interested in moving from mixed-bag network models to compartmentalized network models. However, the number of blocked reactions in single species metabolic models built with AGORA and AGREDA was similar (near 25% of blocked reactions), which illustrates that this problem is not particularly important in AGREDA but in all existing reconstructions.

4. There is unnecessarily detailed information about annotation of SEED and other databases. This can be briefly written and move to a detailed supplementary appendix. However, the most important part is creating the AGREDA model and its usage should be written clearly.

Response: We really appreciate this comment. The methods section of the main manuscript has been amended and new sections included in the Supplementary Materials to fulfill these suggestions made by the reviewer.

5. How is the objective function of AGREDA generated? What is the biomass reaction for FVA?

Response: As noted above, each species contains its own biomass reaction, directly extracted from AGORA. Therefore, we could potentially apply to AGREDA different approaches in the literature to handle this situation, ranging from multi-objective approaches to models where all species grow at the same rate (Magnúsdóttir and Thiele, 2018). Nevertheless, in the present work, we analyzed output metabolites that can be obtained from the degradation of input metabolites presented in different recipes. For this qualitative analysis, we did not consider biomass production. This has been reflected in the Methods section.

Note here that the percentage of maximum biomass to be produced is an optional feature in standard implementations of FVA and, in our analysis, this value was fixed to zero. We checked that the main results presented in the manuscript are not modified by forcing a minimum production of biomass in FVA.

6. Is AGREDA model mass- and energy-balanced?

Response: A few minor corrections have been performed to the AGREDA model presented in the first submission in order to mass-balance the model. We used the function *checkBalance* from the COBRA Toolbox (Heirendt *et al.*, 2019; Vlassis *et al.*, 2014). With respect to energy-balance, we rely on the information provided by metabolic databases used to build AGREDA, namely AGORA (Magnúsdóttir *et al.*, 2017), the Model SEED (Henry *et al.*, 2010), KEGG (Kanehisa and Goto, 2000), BRENDA (Jeske *et al.*, 2019), in order to fix reversible/irreversible reactions. This has also been mentioned in the Methods section.

7. In the core of the used gap filling algorithm, what are the rationale of the values for weights were defined? Were all the included reactions obtained after each iteration compatible with each other, if not how the reactions were chosen?

Response: The manuscript shows the following values for the different weights defined for the gap-filling process:

- 0: Core reactions.
- 0.1: Reaction with taxonomic assignment to species from AGORA.
- 50: Reactions with no taxonomic assignment but functional annotation.
- 100: Reactions with neither taxonomic assignment nor functional annotation.
- 1000: Reactions assigned to plant metabolism.

FastCoreWeighted sets these weights as penalty scores for the given reactions to be included by the gap-filling algorithm in the final model. For example, we prefer those reactions with a 0.1 score to be included in the final model (since they are annotated to AGORA species) to those with a 50 score (which are not annotated to species from AGORA but still have a known functional annotation). The values for these weights, although they follow the aforementioned rationale, have been arbitrarily selected. The resulting solution is acceptable, namely FastCoreWeighted only returned 51 reactions without taxonomic assignment to species in AGORA. In order to avoid pathways with low evidence of occurring in species in AGORA, they were removed from the final model. We have extended the Methods section to clarify this point.

On the other hand, FastCoreWeighted requires all the reactions in the core to be compatible. The explanation for the incompatibilities mentioned in the main text is shown by the following toy example: assume that R1 and R2 are two reversible exchange reactions. R1 is the only reaction that introduces metabolite M1 in the cell and, after M1 is degraded by the cellular metabolism, the resulting metabolite M2 is secreted by R2. On the other hand, if M2 is present in the culture medium, R2 can introduce it in the cell and its degradation product M1 is secreted through R1. In consequence, having R1 and R2 simultaneously active in the input sense is not compatible, but they can operate in such way on their own.

Such incompatibilities arose in the gap filling process and the iterative approach explained in the main text was selected to overcome them. Thus, different reactions were included by FastCoreWeighted in each iteration and, at the end of the process, the union of all the reactions was included in the final model. In this set, some of the reactions may not be compatible, but, as shown above, it is not a problem for the overall model.

8. This work is highly focused on polyphenols and case study is conducted on a polyphenol rich food. A normal diet constitutes numerous bioactive molecules. Is the proposed solution biologically relevant?

Response: The importance of polyphenols in diet is illustrated in Figure 3, which shows that polyphenols take part in very different foods and, therefore, AGREDA is biologically relevant. Certainly, AGREDA could be expanded in the future with other bioactive compounds such as terpenes or saponins. However, as now noted in Figure 2d, AGREDA not only includes polyphenols as input compounds from diet. We enrich other families of nutrients, particularly carbohydrates, carotenoids, fats, minerals, phytosterols, vitamins and xanthines, which are also relevant in different foods.

9. The phenolic compounds added to the model. They are already mentioned in the literature as allergenic components. In other words, the model becomes an allergic-specific model in this respect. Therefore, it is normal for allergic pathways or metabolism to change much more. And it may not be very suitable for lean, obese, and celiac situations.

Response: First, the scientific literature contains many papers describing the anti-allergenic effects of some polyphenols (Aye *et al.*, 2020; Li *et al.*, 2020; Park *et al.*, 2020; Sharma and Naura, 2020). Therefore, phenolics are not listed in the literature as allergenic components, although some could have allergenic effects. On the other hand, phenolic compounds are well known for their health-promoting effects, such as antioxidant (Heleno *et al.*, 2015; Marcelino *et al.*, 2019; Sharma and Naura, 2020) or anti-inflammatory (Marcelino *et al.*, 2019; Park *et al.*, 2020; Sharma and Naura, 2020). In this sense, phenolic compounds, for instance, can play protective roles against different types of cancer (Heleno *et al.*, 2015). However, this does not make the model specific for inflammatory diseases, high oxidative stress situations or cancer. It rather illustrates the importance of considering the metabolism of phenolic compounds in a large variety of health contexts.

10. Several times, AGORA was defined as the largest repository of gut bacterial metabolic reconstructions in the literature. However, Machado et al (reference no 8) automatically created 5587 bacterial models by using the CarveMe tool.

Response: The comment of the reviewer is pertinent. We have rephrased this sentence throughout the manuscript.

11. It is stated that AGREDA is more accurate and sensitive, however, it is not clear what the comparison was made. I suggest the authors perform comparative analysis with AGORA and other models available. It is very well known that AGORA models have major issues which have been discussed by Nielsen's group in a commentary at Nature Biotechnology.

Response: Following the comment of the reviewer, we have included CarveMe in the comparison with AGREDA. In particular, we integrated the 5587 metabolic models reported in CarveMe tool and carried out the following analysis.

1. We identified the number of diet-derived compounds shared by CarveMe and i-Diet, the commercial nutritional software used in our study. We only found 92 out of 650 compounds in CarveMe, which is slightly worse than AGORA (99 out of 650) and substantially worse than AGREDA (330 out of 650). This result is now reflected in the Introduction section.

2. Using the CarveMe tool, we conducted the analysis of nutrient composition for 20 different recipes considered in Figure 3. The comparison of AGREDA and CarveMe can be found in Supplementary Figure 1. Again, AGREDA shows a better performance than CarveMe.

3. Finally, we carried out with CarveMe the analysis of output metabolites from the fermentation of lentils with children's faeces. The results are substantially worse than those of AGORA and AGREDA (see Supplementary Figure 2).

These results reinforce the important contribution brought by AGREDA to existing gut microbiota metabolic models.

12. The model is separated as external and internal, and the borders of each species have been removed. The necessary reactions were selected from the reaction pool. In this case, are the reactions representing each species balanced in themselves? This is important and not mentioned in the text.

Response: This question has been addressed above in question 3.

13. The number of samples for each case is small. The authors should discuss this in the paper carefully.

Response: The reviewer is correct. We have deleted our previous claims about disease-specific metabolic behavior. We agree with the reviewer that the number of samples was not sufficiently large to make such claims.

We included a new section in the manuscript aiming to assess *in-silico* the metabolic interaction between diet and children's microbiota. References to different clinical conditions in this new section aim to illustrate how to exploit the metabolic insights derived from AGREDA in the context of personalized nutrition.

14. Line 176 - they should change the broad term "allergic to foods" to "cow's milk allergic" like they specify in the Methods.

Response: The reviewer is correct. We have changed it throughout the manuscript.

15. Line 189 - Strictly speaking the AGORA network predicts 5 out of 10 measured phenolic compounds with true negatives detected for quercetin and 34dhpgval in the lean subjects.

Response: This sentence has been deleted from the manuscript, since it may lead to misunderstandings, as the one mentioned by the reviewer. We have rephrased this section of the manuscript to clarify that AGREDA substantially overperforms AGORA and CarveME to predict output phenolic compounds obtained from the fermentation of lentils.

16. Line 285 - It would be ideal if this further experimental validation in a larger cohort of children could be performed and included as part of this paper. Three samples in each clinical group is a very low number.

Response: This is not possible at this stage of our project. As noted above, we have deleted our previous claims about disease-specific metabolic behavior. This was substituted by a new section aiming to analyze *in-silico* the interaction between diet and gut microbiota, illustrating the connection of AGREDA to personalized nutrition.

Reviewer #3 (Expertise: Host-microbiome metabolism and metabolic modeling):

This paper is well written and addressed important knowledge gaps in the metabolic pathways of phytochemical conversion in the gut microbiota. However, I am afraid that this paper seems to fail to demonstrate the fidelity and performance of the model (AGREDA) and its genuine connection to personalized diet/nutrition. In addition, I appreciate the use of the authors' own in vitro experimental data in this paper, but I am afraid that the specific methods used for the model validation based on these experimental data do not sound rigorous. The followings are my detailed comments.

Response: We are very grateful for the reviewer's comments. They helped us significantly to improve the manuscript. We think we could satisfactorily address them. See the responses below.

1. The claim of the connection of this work with "personalized diet/nutrition" seems to be a bit over-selling. For example, did this paper recommend any diet or nutrition specific to each donor of the fecal samples in the paper? In my view, the modeling results in this paper are just the identification of sample-specific metabolic pathways, without much specific dietary implications. Therefore, I am afraid that the connection of this paper to personalized diet/nutrition deserves to be mentioned in the Discussion section to a smaller degree, rather than over-emphasized as in the current manuscript like the main theme of the paper. Another issue is that this paper seems to give an impression that their contextualized network of individual fecal samples are disease-specific. However, the number of these samples for each disease group is not large enough to control for any confounding factors in the analysis of the disease group-associated data, and also this paper didn't try to control for these confounding factors. Therefore, I am not convinced about whether their analysis results are really disease-specific or are the reflection of other features of the sample donors.

Response: We have added a new section to the manuscript aiming to address the connection of AGREDA to personalized nutrition: "In-silico prediction of metabolic interaction between diet and children microbiota". In addition, we have modified the Discussion section to envision how AGREDA can help in the development of personalized nutrition programs.

In summary, for the same Spanish children considered in the study of lentil fermentation, AGREDA was used to predict the list of output microbial metabolites that can be potentially obtained from 20 different typical recipes. We could identify output microbial metabolites whose production is linked to the gut microbiota of specific clinical conditions, based on a limited number of samples, but also to specific recipes or to both factors. We obtained a higher number of significant metabolic associations between diet and gut microbiota than AGORA, which illustrates that AGREDA is able to capture diet-specific metabolic pathways that are not present in AGORA. This is important for developing personalized nutrition programs, since the regulation of key output microbial metabolites through diet is an essential part of it.

On the other hand, we have deleted our previous claims about disease-specific metabolic behavior. We agree with the reviewer that the number of samples was not sufficiently large to make such claims. Note here that references to different clinical conditions in the new section of the manuscript aim to illustrate how to exploit the results derived from AGREDA to make connections between gut microbiota and diet.

Finally, following the comment of the reviewer, we have also changed the title of the manuscript by a new one that better reflects the work presented, namely "An extended reconstruction of human gut microbiota metabolism of dietary compounds".

2. The authors justify the use of the supra-organism-like pathways as for the saving of computational costs. However, the supra-organism approach can radically over-simplify the gut community metabolism driven by inter-species metabolic interactions in the gut. Hence, rigorous theoretical justification and detailed assessment of their approach should be provided. Otherwise, even the model outcome is backed-up by some experimental data, this can be potentially "lucky" match between the model results and the presented particular experimental data, rather than indicates robust consistency with experimental data in various settings.

Response: The Methods section of the new version of the manuscript contains a more exhaustive description of the supra-organism strategy, which is now called mixed-bag network strategy, a term introduced in Henry *et al.* (2016), where a detailed assessment and justification of this approach is provided. In that work, authors show that a mixed-bag network strategy produces equal or better results than compartmentalized network approaches. In addition, a toy example has been included in the Supplementary Note 1 to clarify this point.

3. This paper doesn't seem to compute any fluxes in the metabolic pathways using FBA. For example, during the model construction, biomass reactions of the supra-organism network do not seem to be designated for the use in the gap-filling processes. If this gap-filling process was indeed done without any biomass reactions, the gaps would have been filled anyhow without a need for microbial growth. This can be a source of substantial false-positive or false-negative reactions after the gap-filling, because the gap-filling results are likely to be highly-dependent on the specific form of the biomass reactions.

Response: We have expanded the Methods section to address this issue. In the analysis presented in the Results section, for each sample, we assessed which output metabolites can be potentially produced from input compounds available in the different recipes. For this structural (qualitative) study, we used Flux Variability Analysis, which is a standard approach in the field of constraint-based modeling.

On the other hand, AGREDA included a different biomass reaction for each species, which were directly extracted from AGORA. In addition, we guaranteed that biomass production is possible for all species in AGREDA in the gap filling process, because different biomass reactions are part of the required core of FastCoreWeighted. However, our key question in the gap filling process is to connect input dietary compounds with output microbial metabolites or intermediary compounds required from the production of biomass. This has been reflected in the Methods section.

Because the authors did not seem to solve FBA (with specific biomass reactions) when running their constructed model either, the model results are just likely the "network topology"-based ones. Such a pure topology analysis does not really "predict" activated metabolic pathways for a given set of input nutrients. To infer such activated metabolic pathways using the network topology, the contextualization of the network needs to be done (as in this paper) with the substantial amount of empirical data e.g. 16S rRNA gene sequencing data and other omics data. To me, relying on such substantial experimental data itself questions whether this method is indeed "predictive" enough for unknown facts. Moreover, in this case of the contextualization of the supra-organism network topology, I wonder what would be the fundamental difference between the approach in this paper and the previously-established method of the network contextualization such as HUMANN.

Response: We have extended the Methods section to clarify how AGREDA was used to make predictions and its connection to constraint-based models. This has been described in detail in the section "Building context-specific AGREDA and AGORA models". In Systems Biology, there is a clear distinction between topological and constraint-based models, considering the use of

stoichiometry matrix and mass balance. Our approach is not topological because we consider mass balance, as now noted in Methods section (Eq.1).

With respect to HUMAN, our approach is fundamentally different. We used a network-based approach to predict output metabolites from input metabolites from the diet. HUMAN aims to identify annotated metabolic pathways and genes in different conditions but they are not integrated in a network and mass balance is not applied.

Finally, contextualizing based on -omics data is an essential part of most predictive constraint-based models. This is not something specific of AGREDA but pertains to plenty of works in the literature. Our predictions are output microbial metabolites that are derived from the degradation of input dietary compounds. The capacity of contextualizing our model based on the known composition of the gut microbiota of an individual is the aspect that renders our work useful for personalized nutrition and medicine.

Together, Without de novo flux calculation, the use of genome-scale metabolic models is limited to the topology analysis, and hence the full power of the genome-scale metabolic model on the prediction of the flux values was not harnessed in the paper. In other words, the authors' claim on the true value of genome-scale metabolic models (on the personalized nutrition) is not supported enough in the paper.

Response: In our opinion, this point has been addressed in the previous questions. The Methods section has been extended to make clear that we are using genome-scale metabolic models and metabolic flux analysis is conducted through FVA.

4. Transporters in known genome annotations are notoriously poorly annotated, even for primary metabolites. For the plant secondary metabolites focused in this paper, the annotation quality might be even worse. On the other hand, the presence/absence of transporters in an organism is not even always consistent with the true nutrient transport in the microbe, because a number of metabolites can freely diffuse into/out of the cell without the help of the transporters. In the case of phytochemicals focused in this paper, they may also include membrane-diffusible compounds (such as fat-soluble metabolites). These parts are not adequately considered in ModelSEED that the authors used.

Response: Our model does not require the presence of transporters for a given compound to incorporate it into the metabolic reconstruction. In fact, we assume that all dietary compounds can enter the cell and be used as substrates, precisely because of the various reasons correctly pointed out by the reviewer, i.e., the fact that many compounds can enter the cell by passive diffusion, and that many transporters are not yet correctly annotated in current databases.

Actually, it can be potential sources of numerous errors in the model AGREDA in the paper. In addition, I am afraid of other model quality issues as well, including possibly many false-positive reactions, as discussed below. In Line 161, "We can, therefore, conclude that AGREDA provides us with a more accurate tool to assess the effects of the different diets on the gut metabolism with a straightforward application to personalized nutrition." I am afraid that this sentence sounds rather overselling. The main reason of this analysis result (regarding the compounds in different recipes) is that AGREDA includes more pathways for the degradation of many phytochemicals, so AGREDA naturally has more power to differentiate different recipes based on their compositions. However, it does not automatically guarantee that AGREDA is "more accurate tool" (the phrase used above) than AGORA. Inclusion of more pathways may mean the presence of more false-positive pathways, as well.

Response: We meant that AGREDA allows us to more accurately simulate the effect of diet, since it includes a substantially higher number of compounds present in the different recipes. We agree with the reviewer that the word 'accurate' is not adequate at that point of the

manuscript. It was substituted by the word “comprehensive”, which emphasizes that AGREDA provides us with a more complete view of the effect of diet in the gut microbiota.

In addition, I wonder how this result can be “with a straightforward application to personalized nutrition” in the above sentence.

Response: This sentence has been deleted from the manuscript.

Line 189: “the reference (uncontextualized) AGORA network only captures 3 out of 10 measured phenolic compounds, while the reference (uncontextualized) AGREDA network contains all the measured metabolites.” Although AGREDA may have better sensitivity than AGORA due to more compounds/pathways added to AGREDA, there would be higher risk of false positives in AGREDA, as well. This point needs to be addressed. If Line 193 (“AGREDA outperforms AGORA regarding accuracy (75% versus 32,5%)”) intended to address this issue of false positives, the definition of the accuracy here should be clearer. The accuracy should be varied across different samples, but it is not clear what these aggregate numbers (75%, 32.5%) mean. The authors may claim that false-positive issues have already been assessed through their own metabolomic data, like “FP” in Figure 4. However, Figure 4 is only for several enriched metabolites in each sample. and many false-positive results are likely to be present in the list of “all” metabolites predicted by the AGREDA for each sample. It'd better compare the comprehensive list of these predicted metabolites and that of the detected metabolites in the metabolome data for each sample and identify the false positive rates (rather than focusing only on a smaller set of the enriched metabolites).

Response: Clearly, an untargeted HPLC-MS approach would be more informative and complete. However, in our opinion, the comparisons we have performed already provide significant evidence to support the predictions made by AGREDA.

In particular, we have re-written this section to clarify why AGREDA outperforms AGORA. As it is now described in the text:

“Then, we compared the predictive potential of the respective context-specific AGORA and AGREDA models in identifying the presence or absence of 10 output microbial phenolic compounds derived from the fermentation of lentils (Figure 4). For validation purposes, we used targeted metabolomics analysis experiments in three inocula per clinical condition, for a total of 12 samples (see Methods section for details, Supplementary Data 3). Thus, we compared AGORA and AGREDA predictions in 120 cases (12 samples x 10 compounds) (Figure 4a).”

Note here that, in order to facilitate the comparison, we have included now in Figure 4b the confusion matrix for both AGORA and AGREDA, and statistical details:

“As summarized in Figure 4b, we found that AGREDA correctly identifies a large portion of positive cases and, thus, the sensitivity of the AGREDA context-specific models is remarkably higher than that of the AGORA context-specific models: 0.777 versus 0.223, respectively. Despite the fact that the specificity of AGREDA is lower than that of AGORA (0.765 versus 0.941, respectively), AGREDA globally outperforms AGORA, according to Matthews correlation coefficient (MCC): 0.412 vs 0.1434, respectively. Note here that we used MCC because it is more appropriate than other metrics in the situation of class imbalance that we have in this study (Chicco and Jurman, 2020). This difference between AGREDA and AGORA is highly significant (Fisher test p-value: 2.55×10^{-5} vs 0.1892, respectively). We repeated the same analysis for CarveMe; however, it could not predict any of the measured compounds and the results are even poorer than those of AGORA (Supplementary Figure 2).”.

With respect to false positives, following the reviewer comment, we compared the results of AGREDA and AGORA in terms of output microbial metabolites from lentil fermentation with

children's faeces. We found that, on average for each sample, AGREDA predicted 90 output metabolites that are not captured in AGORA. In Figure 4, 6 metabolites out of the 10 metabolites selected were predicted as active in AGREDA but not in AGORA at least once. In these cases, we found 3 false positives out of 60 positive predictions by AGREDA. This is a highly significant result (Hypergeometric test p-value: 2.45e-14), and, in our opinion, it provides evidence to support the predictive capacity of AGREDA. Note here that AGREDA obtained the same result than AGORA for the other 4 metabolites considered in Figure 4.

5. In Line 194, "We, therefore, conclude that the new metabolites and degradation pathways included in AGREDA significantly improve our predictive capacity of gut microbiota metabolism and enable the detection of output metabolites not considered in AGORA." To prove the better "predictive" power of AGREDA than that of AGORA, it is important to validate "output" of the models, as also written in the above sentence. However, it is not clear whether the accuracies have been only calculated for intermediate-to-terminal metabolites in the pathways, rather than the inclusion of input phenolic compounds of which presence may simply dominate the overall accuracy values.

Response: We have re-written this section to clarify the comparison between AGREDA and AGORA. As noted above, we have made clear that the comparison was based on predicting the presence or absence of the 10 output phenolic compounds considered in our study in 12 different samples. Input and intermediate phenolic compounds were not considered in our comparison but only output microbial metabolites.

6. Most of "nutrients" newly added to this model seem plant secondary metabolites. However, these metabolites are not usually called "nutrients", but rather the use of the word phytochemicals is more accurate and clearer. They are classified to neither macronutrient nor micronutrient.

Response: The reviewer is right. They are not usually classified as nutrients but rather bioactive molecules or phytochemicals. However, the nutrient definition is "a compound that is needed to fulfill a function in the organism and needs to be taken from food". This is satisfied by the dietary compounds included in AGREDA.

On the other hand, different bioactive compounds (including phenolic compounds) are also called "non-conventional nutrients" (Abuajah *et al.*, 2015) since they exhibit the capacity to modulate one or more metabolic processes or pathways in the body resulting in some physiological benefits and promotion of well-being. In the case of phenolic compounds, such changes can be mediated through epigenetic modifications (Pan *et al.*, 2013). In addition, under the holobiont point of view, phenolic compounds are also a carbon source for different members of the gut microbiota, so that these molecules are also real nutrients.

Despite the facts explained above, following the reviewer suggestions, in order to avoid misunderstandings, we substituted in the manuscript the word nutrients by dietary compounds.

7. Many of plant secondary metabolites are known to have anti-microbial effects (the authors seem already aware of this fact as Line 137 includes a phrase "they act as a defensive system"). If these metabolites in the paper have such anti-microbial effects (which may dominate their effects as consumable metabolites of the microbes in the paper), then how does the validity of the paper's claim on the efficacy of their models be justified? For example, benzoic acid in Line 140 is used as one of the famous chemical food "preservatives" to suppress microbial growth, to my knowledge.

Response: The comment of the reviewer is very interesting. In fact, we are meant to introduce regulatory relationships between input dietary compounds and microbial abundance in future models. However, this is out of the scope of this work.

8. The word "nutrient degradation" is not clearly defined in the introduction or in the front of the Results in the main text. Some audience may initially wonder whether "nutrient degradation" means the pathway of macromolecular degradation to small molecules (that usually happen outside cell membrane) or the entire biotransformation pathway of nutrients to their terminal downstream metabolites (that involves metabolic reactions inside cells). Even after some audience realize that the nutrients in this paper mainly mean phytochemicals, some of them may still wonder whether the degradation pathways considered in this paper cover the literally only the "breakdown" pathways of these phytochemicals, or the entire biochemical reactions regarding these phytochemicals.

Response: As suggested by the reviewer, we have clearly defined in the front of the Results section the meaning of degradation. In particular, we have added the following sentence in the manuscript:

"We present a new metabolic reconstruction of the human gut microbiota that is focused on covering significant gaps in the degradation pathways of dietary compounds into terminal downstream metabolites."

The followings are relatively minor points:

Line 181: 16S rRNA gene sequencing is able to detect usually genus-level taxa. It should be clarified how the authors mapped these identified genera to strain-level AGORA models considered in AGREDA.

Response: As properly identified by the reviewer, many 16S rRNA gene sequences can only be identified to genus level. However, we were also able to identify a considerable amount of ASVs at the species level. The metabolic capabilities of different organisms within a genus could present large variations and, for this reason, we decided to follow a more conservative approach and excluded taxa at genus level. We only considered the sequenced species-level taxa present in the AGORA models. Note here that the addition of those genera to the AGORA models would not affect the results obtained by our approach, but it would probably increase the number of false positives among the new outcomes. This has been clarified in the Methods section.

Line 49: "Systems Biology" => "systems biology"

Response: Corrected.

Line 61: "For example, AGORA has been already applied to predict dietary supplements for Crohn's disease [15]. Using a similar approach, we predicted the effect of solid diet on the gut microbiota metabolism of infants [16]." ==> The efficacy of the existing genome-scale metabolic models in the design of personalized diets should not be over-emphasized. In the case of Ref. [15], its used genome-scale metabolic models (AGORA) are not yet in the form of realistic platform for the computational simulation of the gut environment. For example, AGORA lacks the species-specific growth rates (e.g. without ATP maintenance energy adjustment) and its simulation platform does not consider characteristic environmental factors in the large intestine such as the nutrient mixing by peristalsis that is known as important for gut microbial dynamics. As demonstrated in Ref. [15] itself, the simulated time period is also too short to realize a biologically meaningful gut microbial dynamics. And the "predicted" diet supplements in Ref. [15] were not experimentally validated. In the case of Ref. [16], the claimed experimental validation is rather limited to a particular metabolite and is viewed as rather weak to guarantee the promising value of the existing genome-scale models or networks in the context of personalized diet/nutrition design.

Response: As suggested by the reviewer, we have lessened the promising value of genome-scale models in personalized nutrition. In particular, we added the following sentence:

“For example, AGORA has been already applied to predict dietary supplements for Crohn’s disease¹⁵. Using a similar approach, we predicted the effect of solid diet on the gut microbiota metabolism of infants¹⁶. Despite these early attempts, genome-scale metabolic models of gut microbiota are still in their infancy and further developments are required to make them into a practical tool for personalized nutrition.”

Line 80, 180: “16S rRNA sequencing”, “16S sequencing”
=> “16S rRNA gene sequencing”. Check the appropriate word usage from Microbiome 3, 31 (2015).

Response: We have corrected the nomenclature throughout the manuscript, following the reference article recommended by the reviewer.

Line 86: “genomics”
=> Shouldn’t this be “metagenomics”?

Response: The reviewer is correct. Amended.

Line 144: “The daily intake of phenolic compounds is rather high”
=> I am not sure whether this sentence can be viewed as correct, because of the amount of plant secondary metabolites is very low in a food compared with macro- or micronutrient contents, although they are contained in many plant-based foods.

Response: As the reviewer mentions, the amount of phenolic compounds in foods is low in comparison with macronutrients. However, the daily intake becomes quite high when considering the whole diet and comparing the intake of phenolic compounds (*i.e.* 1-3 g/day) with the intake of most vitamins and many minerals (in the order of mg or µg/day). Only the intake of Na and K is higher than that of phenolic compounds (in the order of 2-5 g/day). According to Rowland *et al.* (2018), phenolics intake is around 820 mg/day, which the author of this paper considers “relatively high intake levels”. Even more, we have found a mean daily intake of phenolic compounds in Spanish children of 2079 mg/day (Hinojosa-Nogueira *et al.*, 2017), which is even in line with the daily intake of Na. We now state in the manuscript that the daily intake of phenolic compounds is high in comparison to that of most micronutrients and mention our estimation of daily intake in Spanish children in the text.

Line 160: “AGREDA performs better at capturing the potential metabolic differences between the recipes.” => The word choice of “potential metabolic differences” sounds like over-selling, because this analysis was simply the analysis of the compounds in the recipes overlapped with the metabolites in AGORA and AGREDA. The phrase “potential metabolic differences” sounds like something about the simulated model outcomes under these two recipes.

Response: This sentence has been rephrased as recommended by the reviewer. Particularly:

“To illustrate our contribution, we first show that AGREDA provides a more complete connection than AGORA to the nutritional composition of 20 typical recipes of the Mediterranean diet.”

Line 268, 270: “Phenol-Explorer” and “SEED” are written without citation here, but it should be cited.

Response: References to Phenol-Explorer and SEED have been included.

LINE 199: “<=” should be written as math symbol for ‘less than or equal to’. And, it is not clear whether the authors performed FDR correction of the P values.

Response: Symbols have been corrected. We have added a new sub-section in Methods section, called “Statistical Analyses”, where we explain that FDR correction was accomplished.

Line 377: “To avoid false positives, we deleted this subset of reactions”
=> The specific criteria for deleting these reactions should be clearly described in detail.

Response: We have rephased this sentence to clarify this subset of reactions was deleted to avoid the inclusion of reactions with limited evidence in species annotated in AGORA.

References

- Abuajah,C.I. *et al.* (2015) Functional components and medicinal properties of food: a review. *J. Food Sci. Technol.*, **52**, 2522–2529.
- Aye,A. *et al.* (2020) Xanthone suppresses allergic contact dermatitis in vitro and in vivo. *Int. Immunopharmacol.*, **78**, 106061.
- Chicco,D. and Jurman,G. (2020) The advantages of the Matthews correlation coefficient (MCC) over F1 score and accuracy in binary classification evaluation. *BMC Genomics*, **21**, 1–13.
- Heirendt,L. *et al.* (2019) Creation and analysis of biochemical constraint-based models: the COBRA Toolbox v3.0. *Nat. Protoc.*, **8**, 321–324.
- Heleno,S.A. *et al.* (2015) Bioactivity of phenolic acids: Metabolites versus parent compounds: A review. *Food Chem.*, **173**, 501–513.
- Henry,C.S. *et al.* (2010) High-throughput generation, optimization and analysis of genome-scale metabolic models. *Nat. Biotechnol.*, **28**, 977–982.
- Henry,C.S. *et al.* (2016) Microbial Community Metabolic Modeling: A Community Data-Driven Network Reconstruction. *J. Cell. Physiol.*, **231**, 2339–2345.
- Hinojosa-Nogueira,D. *et al.* (2017) New Method To Estimate Total Polyphenol Excretion: Comparison of Fast Blue BB versus Folin-Ciocalteu Performance in Urine. *J. Agric. Food Chem.*, **65**, 4216–4222.
- Jeske,L. *et al.* (2019) BRENDA in 2019: A European ELIXIR core data resource. *Nucleic Acids Res.*, **47**, D542–D549.
- Kanehisa,M. and Goto,S. (2000) KEGG: Kyoto Encyclopedia of Genes and Genomes. *Nucleic Acids Res.*, **28**, 27–30.
- Li,Q. *et al.* (2020) Protocatechuic acid supplement alleviates allergic airway inflammation by inhibiting the IL-4R α -STAT6 and Jagged 1/Jagged2-Notch1/Notch2 pathways in allergic asthmatic mice. *Inflamm. Res. Off. J. Eur. Histamine Res. Soc. ... [et al.]*, **69**, 1027–1037.
- Magnúsdóttir,S. *et al.* (2017) Generation of genome-scale metabolic reconstructions for 773 members of the human gut microbiota. *Nat. Biotechnol.*, **35**, 81–89.
- Magnúsdóttir,S. and Thiele,I. (2018) Modeling metabolism of the human gut microbiome. *Curr. Opin. Biotechnol.*, **51**, 90–96.
- Marcelino,G. *et al.* (2019) Effects of Olive Oil and Its Minor Components on Cardiovascular Diseases, Inflammation, and Gut Microbiota. *Nutrients*, **11**.
- Pan,M.-H. *et al.* (2013) Epigenetic and disease targets by polyphenols. *Curr. Pharm. Des.*, **19**, 6156–6185.
- Park,C.-H. *et al.* (2020) Effects of Apigenin on RBL-2H3, RAW264.7, and HaCaT Cells: Anti-Allergic, Anti-Inflammatory, and Skin-Protective Activities. *Int. J. Mol. Sci.*, **21**.
- Rowland,I. *et al.* (2018) Gut microbiota functions: metabolism of nutrients and other food components. *Eur. J. Nutr.*, **57**, 1–24.
- Sharma,S. and Naura,A.S. (2020) Potential of phytochemicals as immune-regulatory compounds in atopic diseases: A review. *Biochem. Pharmacol.*, **173**, 113790.
- Vlassis,N. *et al.* (2014) Fast Reconstruction of Compact Context-Specific Metabolic Network Models. *PLoS Comput. Biol.*, **10**.

Reviewers' Comments:

Reviewer #1:

Remarks to the Author:

Thank you for responding to my previous queries. There are not further comments to be addressed.

Reviewer #2:

Remarks to the Author:

The authors did a very good job during the revision of the paper. I suggest the publication of the paper in its current form.

Reviewer #3:

Remarks to the Author:

I highly appreciate the authors' considerable efforts and time to address my previous comments. I believe that the manuscript has been indeed improved in its clarity and details. However, I am afraid that the authors' response to my comments actually strengthens some of my fundamental concerns about the manuscript.

The manuscript is about the expansion of a previously-published model (AGORA) to the compound repertoire of mainly phytochemicals or phenolic compounds. To make a significant impact on this field, the quality of the model should be reliable enough, or the methodologies in the work should represent a significant advance. I am afraid that the current work may not meet either of these two expectations.

1. As I previously noted, the predecessor model AGORA is difficult to be viewed as a biologically accurate model, and largely lacks the curation for quantitative predictability for microbial growth and metabolism. At most, AGORA can be viewed as moderately-curved, draft metabolic model. Without significant revision/curation of the original AGORA model itself at the beginning, it is unclear to me how the mere expansion of the model to phytochemical pathways can ensure the model quality at a very fundamental basis.

2. As I noted previously, the supra-organism approach has fundamental limitations in terms of ignoring inter-species metabolic interactions. In response to my comment, the authors defended this approach based on some paper. However, I am afraid that such supra-organism model is fundamentally too far from realistic microbial community metabolism in its rationale, and is not likely to attract the genuine attention of the modeling community in terms of the use of the advanced modeling methodologies. In addition, in this work, the supra-organism model construction itself was also based on previously-published methods, seemingly without much fundamental advance in its idea.

3. Even the incorporation of phytochemical pathways into the model in this work feels questionable to me in its fidelity. In response to my concern about the limitation of the current quality of transporter annotations, the authors wrote that they did not use the transporter information itself. I am afraid that this answer made me rather surprised, because it itself is highly indicative of another source of many false-positive pathways in the model. I wonder why the authors did not devote much efforts on the inference of the "transportability" of individual phytochemicals at least to some degrees, as this part can be the first entry towards the reliable model construction. Also, regarding the gapfilling procedure, the authors answered that they tried to fill the metabolic pathways which can process as many dietary compounds as possible. I rather wonder how this approach can be justified in reality, with which biochemical evidence on its ground.

4. Further contextualization of the AGREDA model based on 16S rRNA gene sequence data also has some issues related to the validation of the "power" of their model. The usefulness of the genome-scale metabolic model can be rationally proved by de novo calculation of metabolic fluxes (and this type of the job was even done in the original paper of AGORA), but I am highly concerned by the fact that the "validation" of the AGREDA model is essentially only based on the contextualized model with 16S rRNA gene sequence data. As the authors responded, the contextualized model still uses stoichiometry and mass balance, but I suppose that major predictive power of this model is likely to come from the inclusion of the 16S rRNA gene sequence data, rather than the mass balance and stoichiometry. It is because 16S rRNA gene sequence data alone have much information on the microbial composition, and it is still unclear to me whether the authors ever demonstrated how critical the mass balance and stoichiometry is to the prediction of their metabolomic data, compared to its counterpart (e.g. simpler, topology-based network with contextualization such as HUMANN). If the predictive power of the AGREDA is not higher enough than its counterpart as above, I am afraid that AGREDA model may be rather viewed as "Rube Goldberg machine" in its nature.

5. Moreover, the contextualization was done only on the species level, but not on the genus level. Although the authors claim that the genus-level contextualization may not make much difference in its results, I am afraid that in many cases, 16S rRNA gene sequence data is more dominated by genera-level taxa, and therefore the species-only contextualization is not likely to be a reliable reflection of the microbial community metabolism in a given sample.

6. Because of many fundamental issues in the rationale of the AGREDA model, I am afraid that the sheer match of some model outcomes with the authors' own metabolomic data alone does not guarantee the robust performance of the model in various settings. One of the most rigorous validation of the model can be possible if the authors validate the "individual" species models separately in a number of growth media. However, the current version of the AGREDA model is based on supra-organism modeling and makes it difficult to validate such individual species-level models separately. However, I stress that the current validation using "aggregate" model output of different organisms cannot be viewed as the model validation strong enough to defend all the potential weakness of the model raised above.

The followings are minor points:

7. Abstract: The compound repertoire expanded by AGREDA is mainly phytochemicals (or phenolic compounds), but I wonder why the current abstract does not include such specific information about the model. I believe that the abstract should become more specific and informative enough about the nature of the model.

8. L256: "in-silico" should be written in italic, throughout the manuscript.

Reviewer #4:

Remarks to the Author:

Based on the comments of Reviewer #3 here are some of my suggestions and comments:

1. As the reviewer noted in the comment, AGORA is a moderately-curated, draft metabolic reconstruction. In the current work the authors have included additional reactions for metabolites in phytochemical pathway, but the authors can test the model for any metabolic tasks defined for gut microbiota. This will be one of the ways to show the model quality. Also, the authors can compare the subsystem distribution for AGORA and AGREDA model and show that both the models are different at subsystem level.

2. The concept of supra-organism models is emerging in the field. I agree with Review #3 that supra-

organism model is fundamentally too far from realistic microbial community metabolism in its rationale, but generating a supra-organism model leads to better understanding of the microbiome on the whole. There has been development in the field for reconstruction and applications of such supra-organism model. In the present study, the authors are focusing on phenolic compounds and how they are transformed by gut microbiota. So, it's justified to work with supra-organism model. The only drawback is that they lose the finer details of which organism plays a predominant role in metabolism of specific metabolites. But they can overcome that challenge by looking into the reactions and map them back to the specific organisms.

3. The concern of quality of transporter annotations is valid. The authors can include transport reactions for the newly included metabolites, if they are not present already. Also, describing the gapfilling procedure in details in the methods section can be helpful for answering the concern raised by the reviewer. Including details of gapfilling will be useful for reproducing the results too.

4. Point 4 by the reviewer is valid. The authors can include de novo calculation of metabolic fluxes and compare their results with AGORA model.

5. I agree with the authors that genus-level contextualization may not bring much difference in their results. The resolution of the 16S rRNA seq data depends on the method and the downstream process used. So, not all 16S rRNA seq data are dominated by genera-level taxa.

6. If the aim of the study was to show how the gut microbiota as a whole differs in various conditions, then using the aggregate model is justified. But this approach will be difficult to validate individual species as the reviewer #3 points out. So, defining the objective of the study will be useful in answering the concern raised by Reviewer #3.

REVIEWER COMMENTS

Reviewer #1 (Remarks to the Author):

Thank you for responding to my previous queries. There are not further comments to be addressed.

Response: We would like to thank the reviewer for her/his previous comments, which helped us much to improve the manuscript.

Reviewer #2 (Remarks to the Author):

The authors did a very good job during the revision of the paper. I suggest the publication of the paper in its current form.

Response: We would like to thank the reviewer for her/his previous comments, which helped us much to improve the manuscript.

Reviewer #3 (Remarks to the Author):

I highly appreciate the authors' considerable efforts and time to address my previous comments. I believe that the manuscript has been indeed improved in its clarity and details. However, I am afraid that the authors' response to my comments actually strengthens some of my fundamental concerns about the manuscript.

The manuscript is about the expansion of a previously-published model (AGORA) to the compound repertoire of mainly phytochemicals or phenolic compounds. To make a significant impact on this field, the quality of the model should be reliable enough, or the methodologies in the work should represent a significant advance. I am afraid that the current work may not meet either of these two expectations.

Response: First, we would like to thank the reviewer for her/his previous comments, which helped us much to improve the manuscript. However, we disagree with the reviewer's opinion. We addressed all the major comments raised by the reviewer in the previous review round and showed the higher quality of AGREDA with respect to other existing metabolic reconstructions of the human gut microbiota.

1. As I previously noted, the predecessor model AGORA is difficult to be viewed as a biologically accurate model, and largely lacks the curation for quantitative predictability for microbial growth and metabolism. At most, AGORA can be viewed as moderately-curated, draft metabolic model. Without significant revision/curation of the original AGORA model itself at the beginning, it is unclear to me how the mere expansion of the model to phytochemical pathways can ensure the model quality at a very fundamental basis.

Response: The previous note of the reviewer was a minor point suggesting to lessen the relevance of genome-scale models in personalized nutrition, in the context of the literature cited in the Introduction section. This comment is fundamentally different.

We agree with the reviewer in that AGORA has substantial room for improvement. However, since its publication, AGORA has received hundreds of citations and has been used to make accurate predictions in very different applications in high-impact journals.

AGREDA has a similar level of curation than that of AGORA in the inherited metabolic tasks. As suggested by Reviewer 4, we have included quality checks for some of these tasks, reported in Magnúsdóttir et al. 2017: aerobic and anaerobic growth in different growth media (Western

diet and a high fiber diet), as well as carbon source uptakes and fermentation product secretions for different species. We found very similar results (Supplementary Figure 1).

Second, AGREDA expands AGORA in order to fill significant gaps in the degradation pathways of input dietary compounds into terminal downstream metabolites. We successfully carried out all quality checks previously suggested by the reviewers, particularly: stoichiometric balancing at the single species level and extension of the metabolomics validation at the community level. As suggested by Reviewer 4, we have reinforced the differences between AGORA and AGREDA with two additional comparisons: metabolic distance between organisms and analysis of metabolic subsystems distribution. In both analyses, the relevance brought by AGREDA is clearly observed (Supplementary Figure 2, Supplementary Table 1).

At the subsystem level, it can be observed that relevant functions of the gut microbiota (not present in AGORA) are recovered. In addition to different families of phenolic compounds, the biosynthesis of carotenoids, such as beta-carotene, is included. Beta-carotene is an important precursor of vitamin A and the gut microbiota has an essential role in the production of this metabolite (Srinivasan and Buys, 2019). We also improve amino acid metabolism in AGREDA, particularly the secretion of citrulline, since the gut is a main source of citrulline in the human body (Curis et al. 2007), as well as alternative pathways for biosynthesis of GABA and other derived compounds (Arakawa et al. 2003), caffeine metabolism and methylxanthine biosynthesis (Farag et al. 2020), among others. We have reflected these ideas in the Results section.

2. As I noted previously, the supra-organism approach has fundamental limitations in terms of ignoring inter-species metabolic interactions. In response to my comment, the authors defended this approach based on some paper. However, I am afraid that such supra-organism model is fundamentally too far from realistic microbial community metabolism in its rationale, and is not likely to attract the genuine attention of the modeling community in terms of the use of the advanced modeling methodologies. In addition, in this work, the supra-organism model construction itself was also based on previously-published methods, seemingly without much fundamental advance in its idea.

Response: Following the very good recommendations of Reviewer 2, AGREDA is now ready to build community models at the species level, as it was previously mentioned in the Discussion section. In fact, we did not include this analysis previously due to its high computation time and the good validation results reached with mixed-bag models.

In order to clarify this issue, we reproduced the case study of lentil fermentation using a compartmentalized network community model and found very similar results (Supplementary Tables 4-5), but at a substantially higher computational expense than that of the mixed-bag network community model (132 minutes vs 5 minutes, respectively). These results reinforce the plausibility of the mixed-bag model adopted.

3. Even the incorporation of phytochemical pathways into the model in this work feels questionable to me in its fidelity. In response to my concern about the limitation of the current quality of transporter annotations, the authors wrote that they did not use the transporter information itself. I am afraid that this answer made me rather surprised, because it itself is highly indicative of another source of many false-positive pathways in the model. I wonder why the authors did not devote much efforts on the inference of the "transportability" of individual phytochemicals at least to some degrees, as this part can be the first entry towards the reliable model construction. Also, regarding the gapfilling procedure, the authors answered that they tried to fill the metabolic pathways which can process as many dietary compounds as possible. I rather wonder how this approach can be justified in reality, with which biochemical evidence on its ground.

Response: In the new version of the manuscript, we have reinforced the idea that we are focused on predicting the metabolic capabilities of the gut microbiota as a whole in different conditions. Since we have checked that all the input dietary metabolites considered in our analysis can be degraded by the gut microbiota, the issue of transport should not be a major source of false positives for the type of analysis conducted here. This is illustrated in the untargeted metabolomics analysis of lentil fermentation with children's faeces. In addition, the transport of input dietary compounds was only included in the species involving their degradation reactions in the metabolic model, which is a conservative strategy. This has been included in Methods section.

With respect to the gap filling process, we have clarified this issue in the Results and Methods section. In particular, we have made clear that all the reactions extracted from the universal database to fill existing gaps included taxonomic assignment to species in AGORA, which provides us with reliable support for the predicted pathways. As described in detail in the Methods section, reactions were annotated to species present in AGORA through their EC numbers using different bioinformatics tools and metabolic databases. We have extended our toy example in Supplementary Note 1 to describe in detail how the gap filling process was performed.

4. Further contextualization of the AGREDA model based on 16S rRNA gene sequence data also has some issues related to the validation of the "power" of their model. The usefulness of the genome-scale metabolic model can be rationally proved by de novo calculation of metabolic fluxes (and this type of the job was even done in the original paper of AGORA), but I am highly concerned by the fact that the "validation" of the AGREDA model is essentially only based on the contextualized model with 16S rRNA gene sequence data. As the authors responded, the contextualized model still uses stoichiometry and mass balance, but I suppose that major predictive power of this model is likely to come from the inclusion of the 16S rRNA gene sequence data, rather than the mass balance and stoichiometry. It is because 16S rRNA gene sequence data alone have much information on the microbial composition, and it is still unclear to me whether the authors ever demonstrated how critical the mass balance and stoichiometry is to the prediction of their metabolomic data, compared to its counterpart (e.g. simpler, topology-based network with contextualization such as HUMANN). If the predictive power of the AGREDA is not higher enough than its counterpart as above, I am afraid that AGREDA model may be rather viewed as "Rube Goldberg machine" in its nature.

Response: First, in the field of systems biology, it has been demonstrated several times that for the analysis of cellular metabolism stoichiometric-based methods are more accurate than graph-based methods even for qualitative (structural) analyses (the type of study conducted here). This is discussed, for instance, in a Bioinformatics article by de Figueiredo et al. 2008. In fact, we used network topology methods in the past and gave them up precisely because of the issue of false positives. Not considering stoichiometry leads to meaningless metabolic pathways and false positives. This is what is shown in the article above and many others. For this reason, we consider unnecessary to demonstrate the relevance of stoichiometry.

Second, with the type of data we have in hand, namely presence/absence of nutrients in different diets and 16S rRNA, we think that a (structural) qualitative analysis is more appropriate. Once we have the amount of nutrients in the different diets, currently under development in the framework of STANCE4HEALTH (<https://www.stance4health.com/>), a European Commission research project about personalized nutrition from which this work is derived, we will be happy to conduct a more quantitative analysis and compare fluxes in different scenarios.

Finally, in order to show that AGREDA can be used to produce quantitative fluxes, we determined aerobic and anaerobic growth rate in different quantitative growth media

reported in Magnúsdóttir et al. 2017, particularly Western diet and a high fiber diet. We obtained very similar flux rates (see Supplementary Figure 1).

5. Moreover, the contextualization was done only on the species level, but not on the genus level. Although the authors claim that the genus-level contextualization may not make much difference in its results, I am afraid that in many cases, 16S rRNA gene sequence data is more dominated by genera-level taxa, and therefore the species-only contextualization is not likely to be a reliable reflection of the microbial community metabolism in a given sample.

Response: In line with Reviewer 4, not all 16S rRNA sequencing data are dominated by genus-level taxa. This is dependent on the dataset and the methodologies employed for analysis and it is not the case for our study. We were able to identify many sequences to species level by using the assignSpecies method within DADA2. This is currently considered the most appropriate method for species-level assignment based on 16S rRNA gene hypervariable regions, based on computational simulations employing a large set of 16S rRNA gene sequences from finished genomes (Edgar et al., 2018). assignSpecies uses exact string matching against the reference database to assign *Genus species* binomials and only ASVs that match a unique reference sequence at 100% identity are assigned at species level rank. In addition, in order not to leave out of our model other species that were highly likely to be present in our dataset, we also included ASVs with 97% identity to species incorporated in AGORA based on MegaBLAST.

We decided to include in our contextualizations only those species for which we could be very certain of their presence in our samples due to the metabolic diversity among different species within a genus. Using genus level assignments for contextualization of the models could result in unreliable metabolic reconstructions, if the actual species in the samples differed substantially in their metabolic capabilities from the ones represented in AGORA.

6. Because of many fundamental issues in the rationale of the AGREDA model, I am afraid that the sheer match of some model outcomes with the authors' own metabolomic data alone does not guarantee the robust performance of the model in various settings. One of the most rigorous validation of the model can be possible if the authors validate the "individual" species models separately in a number of growth media. However, the current version of the AGREDA model is based on supra-organism modeling and makes it difficult to validate such individual species-level models separately. However, I stress that the current validation using "aggregate" model output of different organisms cannot be viewed as the model validation strong enough to defend all the potential weakness of the model raised above.

Response: The reviewer had previously requested completing the validation of our analysis with a more general metabolomics approach, which we performed in the previous round of review. This was her/his previous comment:

“However, Figure 4 is only for several enriched metabolites in each sample. and many false-positive results are likely to be present in the list of "all" metabolites predicted by the AGREDA for each sample. It'd better compare the comprehensive list of these predicted metabolites and that of the detected metabolites in the metabolome data for each sample and identify the false positive rates (rather than focusing only on a smaller set of the enriched metabolites).”

We think that the type of validation conducted is in consonance with the type of analysis performed. We have reinforced throughout the manuscript that the objective of AGREDA is to analyze the metabolic capabilities of the gut microbiota as a whole in different conditions.

The followings are minor points:

7. Abstract: The compound repertoire expanded by AGREDA is mainly phytochemicals (or phenolic compounds), but I wonder why the current abstract does not include such specific information about the model. I believe that the abstract should become more specific and informative enough about the nature of the model.

Response: We have included a sentence in the abstract to clarify that the improvement of AGREDA is mainly focused on phenolic compounds.

8. L256: "in-silico" should be written in italic, throughout the manuscript.

Response: Thanks. Updated.

Reviewer #4 (Remarks to the Author):

Based on the comments of Reviewer #3 here are some of my suggestions and comments:

Response: First, we would like to thank the reviewer for her/his comments on the manuscript. They have helped us much to improve the quality of the manuscript.

1. As the reviewer noted in the comment, AGORA is a moderately-curated, draft metabolic reconstruction. In the current work the authors have included additional reactions for metabolites in phytochemical pathway, but the authors can test the model for any metabolic tasks defined for gut microbiota. This will be one of the ways to show the model quality. Also, the authors can compare the subsystem distribution for AGORA and AGREDA model and show that both the models are different at subsystem level.

Response: Following the reviewer's recommendation, we have now included additional quality checks for other metabolic tasks, particularly those reported in AGORA: aerobic and anaerobic growth in different growth media (Western diet and a high fiber diet), as well as carbon source uptakes and fermentation product secretions for different species. The results of AGREDA are in consonance with the results obtained in AGORA.

In addition, we have reinforced the differences between AGREDA and AGORA with two additional comparisons: metabolic distance between organisms and analysis of metabolic subsystems distribution. In both analyses, the relevance brought by AGREDA is clearly observed (Supplementary Figure 2, Supplementary Table 1).

As noted above, in the subsystem analysis, it can be observed that relevant functions of the gut microbiota (not present in AGORA) are completed. In addition to different families of phenolic compounds, the biosynthesis of carotenoids, such as beta-carotene, is included. Beta-carotene is an important precursor of vitamin A and the gut microbiota has an essential role in the production of this metabolite (Srinivasan and Buys, 2019). We also improve amino acid metabolism in AGREDA, particularly the secretion of citrulline, since the gut is the main source of citrulline in the human body (Curis et al. 2007), as well as alternative pathways for biosynthesis of GABA and other derived compounds (Arakawa et al. 2003), caffeine metabolism and methylxanthine biosynthesis (Farag et al. 2020), among others. We have reflected these ideas in the Results section.

2. The concept of supra-organism models is emerging in the field. I agree with Review #3 that supra-organism model is fundamentally too far from realistic microbial community metabolism in its rationale, but generating a supra-organism model leads to better understanding of the microbiome on the whole. There has been development in the field for reconstruction and applications of such supra-organism model. In the present study, the authors are focusing on phenolic compounds and how they are transformed by gut microbiota. So, it's justified to work with supra-organism model. The only drawback is that they lose the finer details of which organism plays a predominant role in metabolism of specific metabolites. But they can overcome that challenge by looking into the reactions and map them back to the specific organisms.

Response: We agree with the reviewer. However, in order to clarify this issue, we reproduced the case study of lentil fermentation using a compartmentalized network community model and found very similar results (Supplementary Figure 3, Supplementary Tables 4-5), but at a substantially higher computational expense than that of the mixed-bag network community model (132 minutes vs 5 minutes, respectively). These results reinforce the plausibility of the mixed-bag model adopted.

3. The concern of quality of transporter annotations is valid. The authors can include transport reactions for the newly included metabolites, if they are not present already. Also, describing the gapfilling procedure in details in the methods section can be helpful for answering the concern raised by the reviewer. Including details of gapfilling will be useful for reproducing the results too.

Response: Certainly, we had included a transport reaction for the newly included metabolites. However, the transport reaction of an input metabolite was only included in the species involving its degradation reactions in their metabolic model, which is a conservative strategy. This has been included in the Methods section.

With respect to the gap filling process, we have clarified this issue in the Results and Methods section. In particular, we have made clear that all the reactions extracted from the universal database to fill existing gaps included taxonomic assignment to species in AGORA, which provides us with a reliable support for the predicted pathways. As described in detail in the Methods section, reactions were annotated to species present in AGORA through their EC numbers using different bioinformatics tools and metabolic databases. We have extended our toy example in Supplementary Note 1 to describe in detail how the gap filling process was performed.

4. Point 4 by the reviewer is valid. The authors can include de novo calculation of metabolic fluxes and compare their results with AGORA model.

Response: As explained above, with the type of data we have in hand, namely presence/absence of nutrients in different diets and 16S rRNA, we think that a (structural) qualitative analysis is more appropriate. Once we have the amount of nutrients in the different diets, currently under development in the framework of STANCE4HEALTH (<https://www.stance4health.com/>), a European Commission research project about personalized nutrition from which this work is derived, we will be happy to conduct a more quantitative analysis and compare fluxes in different scenarios. In other words, the calculation of fluxes with the data available will not provide additional insights to the ones currently reported in the manuscript.

In order to show that AGREDA can be used to produce quantitative fluxes, we determined aerobic and anaerobic growth rate in different quantitative growth media reported in Magnúsdóttir et al. 2017, particularly Western diet and a high fiber diet. We obtained very similar flux rates (see Supplementary Figure 1).

5. I agree with the authors that genus-level contextualization may not bring much difference in their results. The resolution of the 16S rRNA seq data depends on the method and the downstream process used. So, not all 16S rRNA seq data are dominated by genera-level taxa.

Response: Thanks. See above our detailed response to Reviewer 3.

6. If the aim of the study was to show how the gut microbiota as a whole differs in various conditions, then using the aggregate model is justified. But this approach will be difficult to validate individual species as the reviewer #3 points out. So, defining the objective of the study will be useful in answering the concern raised by Reviewer #3.

Response: Thanks for the recommendation. We also think that the type of validation conducted is in consonance with the type of analysis performed. We have reinforced throughout the manuscript that the objective of AGREDA is to analyze in different conditions the metabolic capabilities of the gut microbiota as a whole.

References

Curis, E., Crenn, P., & Cynober, L. (2007). Citrulline and the gut. *Current Opinion in Clinical Nutrition & Metabolic Care*, 10(5), 620-626.

Edgar, R. C. (2018) Updating the 97% identity threshold for 16S ribosomal RNA OTUs. *Bioinformatics*, 34(14), 2371–2375

Farag, M. A., Abdelwareth, A., Sallam, I. E., El Shorbagi, M., Jehmlich, N., Fritz-Wallace, K., ... & von Bergen, M. (2020). Metabolomics reveals impact of seven functional foods on metabolic pathways in a gut microbiota model. *Journal of advanced research*, 23, 47-59.

Magnúsdóttir, S., Heinken, A., Kutt, L., Ravcheev, D. A., Bauer, E., Noronha, A., ... & Thiele, I. (2017). Generation of genome-scale metabolic reconstructions for 773 members of the human gut microbiota. *Nature biotechnology*, 35(1), 81-89.

Srinivasan, Krishnamoorthy, and Elna M. Buys. "Insights into the role of bacteria in vitamin A biosynthesis: Future research opportunities." *Critical reviews in food science and nutrition* 59.19 (2019): 3211-3226.

Yorifuji, T., Shimizu, E., Hirata, H., Imada, K., Katsumi, T., & Sawamura, S. I. (1992). Guanidinobutyrase for L-arginine degradation in *Brevibacterium helvolum*. *Bioscience, biotechnology, and biochemistry*, 56(5), 773-777.

Reviewers' Comments:

Reviewer #4:

Remarks to the Author:

The authors have done a great work in editing the manuscript and answering all the comments. I have a minor comment, in the rebuttal letter the authors mention that the computational expense was "132 minutes vs 5 minutes" for the mixed-bag network community model, but in the manuscript it is written as "120 minutes vs 5 minutes". Kindly edit the sentence accordingly. Other than that, I found the work to be very interesting and the manuscript has improved a lot upon revision.

REVIEWER COMMENTS

Reviewer #4 (Remarks to the Author):

The authors have done a great work in editing the manuscript and answering all the comments. I have a minor comment, in the rebuttal letter the authors mention that the computational expense was "132 minutes vs 5 minutes" for the mixed-bag network community model, but in the manuscript it is written as "120 minutes vs 5 minutes". Kindly edit the sentence accordingly. Other than that, I found the work to be very interesting and the manuscript has improved a lot upon revision.

Response: The correct computational expense is "132 minutes vs. 5 minutes". We have amended the text in the main manuscript accordingly.

We would like to thank the reviewers for the thorough revision of our work. We believe that their comments have substantially improved the quality of our article.